# Altered visual cortex excitability in premenstrual dysphoric disorder: Evidence from magnetoencephalographic gamma oscillations and perceptual suppression

**Viktoriya O. Manyukhina**[1,2], **Elena V. Orekhova**[1]*, **Andrey O. Prokofyev**[1], **Tatiana S. Obukhova**[1], **Tatiana A. Stroganova**[1]

**1** Center for Neurocognitive Research (MEG Center), Moscow State University of Psychology and Education, Moscow, Russian Federation, **2** National Research University Higher School of Economics, Moscow, Russian Federation

* orekhova.elena.v@gmail.com

**Data Availability Statement:** All relevant data are available in the Supporting Information files.

## Abstract

Premenstrual dysphoric disorder (PMDD) is a psychiatric condition characterized by extreme mood shifts during the luteal phase of the menstrual cycle (MC) due to abnormal sensitivity to neurosteroids and unbalanced neural excitation/inhibition (E/I) ratio. We hypothesized that in women with PMDD in the luteal phase, these factors would alter the frequency of magnetoencephalographic visual gamma oscillations, affect modulation of their power by excitatory drive, and decrease perceptual spatial suppression. Women with PMDD and control women were examined twice–during the follicular and luteal phases of their MC. We recorded visual gamma response (GR) while modulating the excitatory drive by increasing the drift rate of the high-contrast grating (static, 'slow', 'medium', and 'fast'). Contrary to our expectations, GR frequency was not affected in women with PMDD in either phase of the MC. GR power suppression, which is normally associated with a switch from the 'optimal' for GR slow drift rate to the medium drift rate, was reduced in women with PMDD and was the only GR parameter that distinguished them from control participants specifically in the luteal phase and predicted severity of their premenstrual symptoms. Over and above the atypical luteal GR suppression, in both phases of the MC women with PMDD had abnormally strong GR facilitation caused by a switch from the 'suboptimal' static to the 'optimal' slow drift rate. Perceptual spatial suppression did not differ between the groups but decreased from the follicular to the luteal phase only in PMDD women. The atypical modulation of GR power suggests that neuronal excitability in the visual cortex is constitutively elevated in PMDD and that this E/I imbalance is further exacerbated during the luteal phase. However, the unaltered GR frequency does not support the hypothesis of inhibitory neuron dysfunction in PMDD.

**Funding:** This study was funded by Moscow State University of Psychology and Education (MSUPE). The funder had no role in study design, data collection and analysis, decision to publish, or preparation of the manuscript.

**Competing interests:** The authors have declared that no competing interests exist.

# Introduction

Female steroid hormones not only support reproduction but also regulate excitability of the neural cells. These hormones enter the brain by crossing the blood-brain barrier or are synthesized in the brain [1–3] and affect emotions, perception, and memory [4]. In most females of reproductive age hormone fluctuations during the menstrual cycle (MC) cause only mild changes in physical state and mood a few days before menstruation, known as premenstrual symptoms (PMS) [5, 6]. However, in 2–5% of women, these fluctuations result in disabling psychiatric problems, also known as premenstrual dysphoric disorder (PMDD) [7, 8]. Over the past decades, PMDD has attracted the attention of both clinicians and the scientific community [9], as it is associated with substantial disruption of women's everyday activities, inability to work and high risks of suicide [10–12].

Normal as well as pathological changes in the brain during the MC to a large extent depend on the varying concentration of two steroid hormones–progesterone and estradiol. Although the mechanisms of their action are not fully understood, there is evidence that progesterone and estradiol either directly or indirectly modulate neuronal excitability and affect excitation-to-inhibition (E/I) balance in the brain [13].

There is evidence that PMDD symptoms are related to an altered neural inhibition which results from abnormal sensitivity to neurosteroids rather than their altered levels in the brain [14–18]. The neuroactive effects of progesterone associated with PMS are thought to be mediated by its neurosteroid metabolite allopregnanolone (ALLO) [14, 16, 19]. ALLO is a positive allosteric modulator of GABA$_A$ receptors (GABA$_A$Rs) that primarily facilitates tonic inhibition mediated by δ subunit-containing GABA$_A$Rs (δGABA$_A$Rs) [20]. However, under certain conditions, it can also produce a paradoxical excitatory effect [21–24]. Estradiol also affects neuronal excitability via several pathways [25–32]. In general, both progesterone and estradiol produce complex effects on neuronal excitability that depend on the brain area, neurosteroid levels, and other factors.

Symptoms of PMDD, such as depressed mood, anger, irritability, sleep problems, etc. appear to be primarily associated with functional abnormalities in the brain regions involved in emotional regulation [33–35]. However, changes associated with PMDD are not limited to these regions [36–40]. In particular, the activity of the visual cortex is affected by neurosteroids [41–44] and may reflect MC-related abnormalities in the E/I balance regulation in women with PMDD. Primary cortical areas, including visual cortex, display high concentrations of GABAergic receptors [45] and neurons [46] and their GABAergic tone is sensitive to fluctuations in ALLO [44]. Estradiol is another potent modulator of visual cortex excitability [41].

Evidence for the sensitivity of neural activity in the visual cortex to steroid hormones during the MC also comes from several electrophysiological studies. Amplitude and latency of cortical visual evoked potentials [47–50], as well as magnitude of the transcranial magnetic stimulation (TMS) paired-pulse suppression of visual evoked potentials [43] change with phase of the MC. Parameters of the visual alpha rhythm in women without severe PMS are also affected by phase of the MC [51–54]. Besides, in a recent study Sumner and colleagues have shown that frequency of gamma oscillations induced by static and moving visual gratings also changes during the MC in women without severe PMS [55].

Functional changes in the visual cortex during the MC may reflect modulatory effects of neurosteroids on excitatory and inhibitory neurotransmission. Gamma oscillations may be particularly sensitive to these modulatory effects, as they closely reflect local cortical interaction of excitatory and inhibitory neurons and are sensitive to the balance of their activity (E/I balance) [56–58]. These oscillations are effectively induced by high-contrast regular patterns,

such as gratings, and can be reliably recorded in humans using magnetoencephalography (MEG) [59, 60].

Our previous MEG studies suggest that individual variations in regulation of the E/I balance are better captured by stimulation-related *changes* of oscillatory gamma response (GR) rather than by GR amplitude or frequency measured in a single experimental condition [61–63]. Strong and predictable modulations of GR power and frequency can be achieved when modulating the excitatory drive to the visual cortex by increasing drift rate of a large high contrast visual grating: while the frequency of gamma oscillations increases nearly linearly with increasing drift rate, the GR power usually increases with transition from static to slowly drifting grating and then decreases with further increase of the drift rate. As we discussed elsewhere [62–65], this pattern of changes in GR power may reflect efficiency of neuronal inhibition: a certain level of activity of inhibitory interneurons is necessary for synchronization of gamma oscillations [66, 67], but their 'too strong' activation–which in our experimental paradigm was associated with 'medium' and 'fast' drift rates [65]–may lead to a disruption of synchronization between principal cells and interneurons resulting in reduced amplitude of gamma oscillations [68, 69]. Following this logic, a higher neural E/I ratio would lead to less suppression of GR power for the same level of excitatory drive (i.e., at the same drift rate of a grating), because in this case more inhibition is required to desynchronize strongly excited principal cells [68]. We therefore predicted that the presumably elevated E/I ratio in the visual cortex in women with PMDD during the symptomatic premenstrual period would result in relatively weaker GR power suppression with increasing drift rate.

Next, we predicted that MC phase-related changes in E/I ratio in women with PMDD would affect the frequency of their gamma oscillations. Animal studies suggest that gamma frequency strongly depends on the tonic excitability of inhibitory neurons, which is regulated through $\delta GABA_A Rs$, so that the lower the tonic inhibition of the inhibitory interneurons, the higher the gamma frequency [66, 70]. The $\delta GABA_A Rs$ are highly sensitive to neurosteroids, and their numbers increase following an increase in progesterone levels [71–73]. However, Sumner et al. [55] found that in women without severe PMS, the frequency of visual gamma oscillations was *higher* during high-progesterone-and-ALLO luteal phase than during low-progesterone-and-ALLO follicular phase. The authors hypothesized that these changes might be caused by developing tolerance to ALLO during late luteal phase. Since abnormal sensitivity to ALLO has previously been suggested to be one of the mechanisms of PMDD [18], we hypothesized that it may manifest as a change in GR frequency during the luteal phase.

Thirdly, we expected that abnormal neural inhibition in women with PMDD during their symptomatic luteal phase would affect visual perception functions that are highly sensitive to the strength of neural inhibition in the visual cortex. To test this prediction, we assessed in our participants perceptual spatial suppression–the capacity of the visual system to suppress the perception of large background-like motion [74]. Spatial suppression is reflected in an increase in the time it takes to discriminate the direction of motion of regularly patterned visual objects as their size increases [75, 76]. This perceptual phenomenon can be explained by an increase of surround inhibition associated with stimulation of the 'far surround' of the neurons' receptive fields [77, 78]. Indeed, elderly people [79, 80] or individuals with disorders associated with weakened neural inhibition [81–84] demonstrated decreased spatial suppression.

Herein, to test these predictions, we investigated 1) visual MEG gamma oscillations and 2) perceptual spatial suppression in women with PMDD and women without severe PMS during the early-to-mid follicular and the mid-to-late luteal phases of the MC, when the PMS are least and most pronounced, respectively. For this purpose, we invited the same women twice, counterbalancing the phases of the first visit (follicular and luteal) between the participants.

## Materials & methods

### Participants

The participants from the PMDD group were recruited via media advertisements (social networks and publications in the media). The control women were recruited among members of the internet 'healthy life style' group and students. The criteria for exclusion, common for both groups, included (a) pregnancy or lactation, (b) irregular MC, (c) intake of hormones (contraceptive pills, thyroid hormones, etc.) or psychoactive drugs (nootropics, antidepressants, tranquilisers, etc.). Control women also had to have no personal or family history of neurological or psychiatric disorders. Women with PMDD who had been diagnosed with depression at some point in their lives (two of twenty women) were still included in the study because PMDD is often comorbid with this psychiatric condition [85–87].

The potential subjects of the clinical group reported severe changes in mood associated with the MC, which had a significant impact on their lives, relationships, daily activities, and job. To test whether these symptoms met criteria for PMDD, we used Carolina Premenstrual Assessment Scoring System (C-PASS, [88]). The C-PASS measures severity of PMS according to DSM-5 PMDD criteria [89]. It includes 20 statements related to 11 symptoms, which should be rated from 1 ('not at all') to 6 ('extreme symptoms') according to their severity. All volunteers of the PMDD group were asked to fill in the C-PASS scales daily for two or three MCs. They were included in the study only if they met the C-PASS criteria for PMDD in at least two MCs. All but two PMDD subjects continued to fill in the C-PASS during the period when the MEG and psychophysical investigation were performed, which allowed us to assess the severity of their PMDD symptoms on the day of examination.

To estimate the symptom severity on the day of examination (hereafter 'same-day PMS scores'), we averaged scores obtained on this day for the symptoms that were reliably associated with PMDD according to the C-PASS (i.e., the symptoms which were strong enough and varied during the MC; see [88]). The same-day PMS scores were available in 18 out of 20 participants with PMDD.

We expected that using the C-PASS among control participants could greatly increase the dropout. It could also lead to a preponderance of control volunteers with certain personality traits (e.g., those who are persistent enough to follow the requirements without being interested in the C-PASS results). Therefore, to ensure that control participants did not have severe PMS, we used a less time-consuming premenstrual tension syndrome scale (PMTS-VAS, [90]). In PMTS-VAS, subjects need to retrospectively assess their symptoms on a visual analogue scale for one week after the start of their last menstruation (early follicular phase) and for a week before their last menstruation (late luteal phase), when the PMS are expected to be least and most pronounced, respectively. PMTS-VAS consists of 12 statements that satisfy the DSM-4 criteria for PMDD and should be rated from 0 ('not at all') to 100 ('extreme symptoms') across a continuum of values. In each control participant, we estimated PMS score as a difference between PMTS-VAS scores for the luteal and follicular phases [90, 91]. The resulting PMS score could thus vary from 0 (no PMS) to 100 (maximal symptom severity). In control subjects, the resulting PMS score averaged over 12 symptoms varied from 0 to 33.3 (mean ± standard deviation (S.D.): 16.59 ± 10.41), and PMS score averaged over core mood symptoms (depression, anxiety, irritability, and lability of mood) varied from 0 to 37.5 (mean ± S.D.: 18.53 ± 13.87), which is below the cut-off for severe PMS according to Steiner et al. [91].

Neurophysiological (MEG) and behavioral (psychophysical) data were collected from 20 women who fulfilled the criteria for PMDD (PMDD group) and 27 age-matched control women. PMDD and control subjects did not differ in age (Student's t-test, t(45) = 0.81,

**Table 1. Characteristics of the experimental groups.**

| Characteristic | PMDD group (N = 20) Mean (S.D.) [range] | Control group (N = 27) Mean (S.D.) [range] | Group difference | |
|---|---|---|---|---|
| | | | Student's T | P-value |
| Age (years) | 29.05 (5.38) [18–39] | 27.70 (5.82) [18–40] | 0.81 | 0.42 |
| Average reported length of MC | 28.50 (2.74) [24–34] | 28.48 (2.01) [25–33] | 0.03 | 0.98 |
| Day of MC, follicular phase visit | 5.35 (2.56) [2–10] | 4.81 (2.08) [2–10] | 0.79 | 0.43 |
| Day of MC, luteal phase visit | 23.30 (2.79) [18–30] | 23.33 (2.22) [20–29] | -0.05 | 0.96 |
| Luteal phase visit: days before the next menstruation | 5.60 (2.72) [1–10] | | | |

N–number of subjects; S.D.—standard deviation; MC–menstrual cycle.

p = 0.42), the average length of MC (Student's t-test, t(45) = 0.03, p = 0.98) or day of MC during the visit (follicular: Mann-Whitney U = 241.0, p = 0.27; luteal: Mann-Whitney U = 267.0, p = 0.48). The day of the MC normalized by the average MC length also did not differ between the groups (p's>0.26). For PMDD participants, we also got information about the day when their next menstruation started after their 'luteal phase visit'. Detailed information about the participants is summarized in Table 1.

To test whether subjects from the two groups differed in their level of trait anxiety and their level of anxiety during the laboratory visit, participants were asked to complete the State-Trait Anxiety Inventory (STAI) questionnaire [92] right before the MEG experiment. Additionally, participants with PMDD filled in the screening tests for bipolar disorder (Bipolar Spectrum Diagnostic Scale, BSDS [93]; Mood Disorder Questionnaire, MDQ, [94]; Hypomania Check List, HCL-32 [95]), and for depressive disorder (Beck's Depression Inventory, BDI [96]). The bipolar disorder questionnaires were completed during the asymptomatic follicular phase, and the BDI was completed twice: during the follicular and symptomatic late luteal phases.

All the participants got blood tests for hormones and underwent MEG recording and psychophysical testing on the same day, once during the early-to-mid follicular (hereafter 'follicular') and once during mid-to-late luteal (hereafter 'luteal') phases of the MC. The phase of the 1st visit to the laboratory (follicular or luteal) was counterbalanced between participants. The study has been approved by the Ethical Committee of the Moscow State University of Psychology and Education. Subjects were not paid for their participation. All subjects gave written informed consent.

## Blood samples

The blood samples used to confirm the phase of the MC and to analyze the effect of steroid hormone levels on visual gamma oscillations were collected in the morning. Plasma levels of estradiol and progesterone were measured using a solid phase chemiluminescence immunoassay. The analyzes were performed by INVITRO, a certified commercial laboratory that performs a wide range of medical tests for clinical and diagnostic purposes (https://www.invitro.ru).

## MEG paradigm

Participants watched a sequence of large high-contrast circular gratings (18°, 100% contrast, spatial frequency 1.66 cycles per degree) that either drifted with one of three velocities ('slow': 1.2°/s, 'medium': 3.6°/s, 'fast': 6.0°/s) or remained static (see Fig 1, upper panel). The presentation time for each stimulus ranged randomly from 1.2 to 3 seconds. After this period, the moving grating stopped or the static grating began to move. Participants were instructed to press a button as soon as this change occurred. Each new trial started with a fixed 1.2 s prestimulus

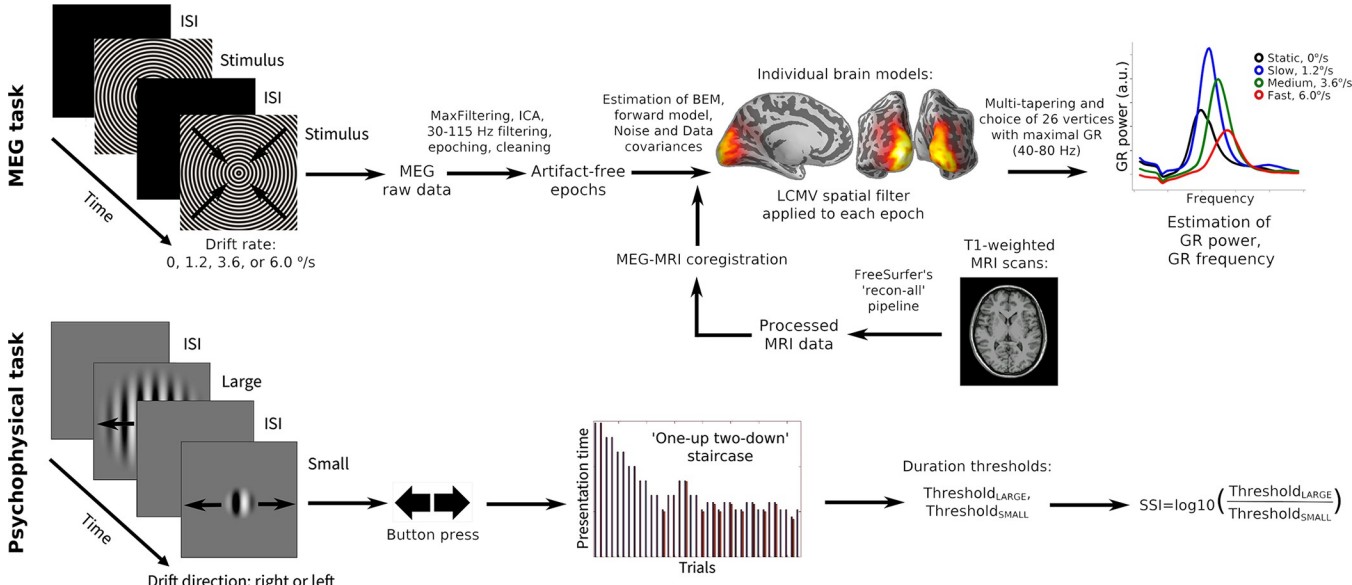

**Fig 1.** Schematic representation of experimental procedures and analysis of MEG (upper panel) and psychophysical data (lower panel). For a detailed description of the experimental procedures and analysis pipelines, see the Methods section. The procedures were identical between the two visits. ISI–inter-stimulus interval.

interval, continued with the presentation of the stimulus, and ended immediately after the button press. To reduce fatigue and boredom, participants were shown short (3–6 s) cartoon animations after every 2–5 gratings. For each participant, 90 gratings of each type were randomized and presented in three sessions. For additional details, see Orekhova et al. [62, 64].

## MRI data acquisition and processing

All participants underwent magnetic resonance imaging (MRI; 1.5 T, voxel size 1 mm x 1 mm x 1 mm). The T1-weighted images were processed with the FreeSurfer's (v.6.0.0) default 'recon-all' pipeline. Its main included steps were motion correction, spatial normalization, skull stripping, gray/white matter segmentation.

## MEG data acquisition, preprocessing, and analysis

MEG data were acquired at the Center for Neurocognitive Research (MEG-center) of the Moscow State University of Psychology and Education using an Electa VectorView Neuromag 306-channel MEG detector array (Helsinki, Finland) consisting of 102 magnetometers and 204 planar gradiometers.

The MEG signal was registered with 0.03 Hz high-pass and 300 Hz low-pass in-built filters and sampled at 1000 Hz. Temporal signal-space separation method (tSSS) [97], a temporal extension of SSS, with 0.9 correlation limit and with movement compensation implemented in MaxFilter software (v.2.2), was applied to raw MEG signal in order to reduce interference of external artifact sources and compensate for movement-related distortions.

All subsequent processing steps were performed using MNE python software (v.0.22.0). Raw data were down-sampled to 500 Hz and independent component analysis (ICA) was applied in order to detect and remove ICs that corresponded to biological artifacts (eye blinks and heartbeats). The number of removed ICs did not differ between the groups and/or phases of the MC (all p's>0.08). The data were then filtered using 30 Hz high-pass and 115 Hz low-

pass finite impulse response filter with default parameters (symmetric, with Hamming window; the window length and width of the transition band at the low/high cut-off was set to 'auto') and epoched from -1 s to 1.2 s relative to the stimulus onset. Epochs contaminated by instrumental and myogenic artifacts were excluded based on visual inspection of unfiltered epochs. The number of resulting artifact-free epochs did not differ significantly between the groups for any of the four velocity conditions and/or phases of the MC (all p's>0.19). Average number of artifact-free epochs for 'static', 'slow', 'medium', and 'fast' conditions in the control group was 83.41, 83.56, 83.11, 83.56 in the follicular phase vs. 81.78, 82.78, 81.62, 82.52 in the luteal phase; and in the PMDD group was 84.05, 84.25, 84.40, 83.40 in the follicular phase vs. 83.35, 83.55, 82.70, 82.85 in the luteal phase.

## MEG source analysis

MEG data were co-registered with the structural MRI of the head, and a single-layer boundary element model was constructed. We then created surface-based source space with 4096 vertex sources in each hemisphere and estimated the forward solution. Noise covariance matrices and data covariance matrices were estimated based on the prestimulus (-1–0 s, relative to the stimulus onset) and stimulation (0–1.2 s, relative to the stimulus onset) time intervals, respectively, of all conditions combined. Linearly constrained minimum variance (LCMV; [98]) beamformer spatial filters with a source orientation that maximizes power and a regularization coefficient 0.05 were created separately for each stimulus type and were applied individually to each epoch. As we expected to observe gamma oscillations in the visual cortex [59, 62–65], the inverse solution was limited to visual and adjacent cortical areas, same as in Manyukhina et al. [61].

## Time-frequency analysis of the MEG data at the source level

The approach to the analysis of visual gamma parameters used in the present study is described in detail by Manyukhina et al. [61]. The time-frequency analysis was performed for the signal at each vertex source using multi-taper method, separately for the four velocity conditions and for the prestimulus and stimulation time intervals. The following parameters were used: bandwidth = 10 Hz, frequency resolution ~ 1.11 Hz, time step = 2 ms. Then, at each vertex source we estimated the normalized response power as (Stim–PreStim)/PreStim, where PreStim and Stim are the spectral powers estimated at -0.9–0 s and 0.3–1.2 s time intervals relative to the stimulus onset, respectively. Among all the four conditions, we then selected individually for each participant the 26 vertices with maximum response power in gamma range (40–80 Hz), and averaged the spectra over these vertices, separately for each condition. Visual inspection of localization results has shown that for all participants the selected vertices were spatially contiguous in the surface mesh of each hemisphere. See Fig 1 (upper panel) for a brief review of MEG data analysis steps.

For each velocity condition, the GR power was calculated as the average of those spectrum values that exceeded 2/3 of the peak value in the frequency range 35–90 Hz. The GR frequency was estimated as a center of gravity of the spectrum values used to calculate the GR power. The GR was considered reliable if, according to Wilcoxon signed-rank test, the probability of no difference in the frequency corresponding to the peak gamma power in prestimulus and stimulation time intervals was below p = 0.0001 (see, e.g., [63, 64]). In all our participants and velocity conditions, the GRs met this reliability criterion.

**Psychophysical perceptual spatial suppression test.** To assess perceptual suppression associated with increasing stimulus size, we used a modification of the spatial suppression paradigm suggested by Tadin and colleagues [75, 76]. The similar experimental paradigm was used in our previous studies [63, 99].

Visual stimuli were presented using PsychoToolbox for Matlab (MathWorks). Subjects sat at 60 cm distance from the monitor (Benq XL2420T, 24″W LED, 1920 × 1080 resolution, 120 Hz). Before the testing, all participants completed a training session.

Inter-trial interval was 500 ms. In the beginning of each trial a central dot flickered at the screen (50 ms on, 50 ms off, 250 ms on, 150 ms off) followed by the stimulus presentation. The stimuli were 12˚, 2.5˚, and 1˚ vertical high-contrast sinusoidal gratings moving at a constant rate of 4˚/s in each trial (Fig 1, lower panel). Direction of motion (left or right) was determined randomly for each trial. The participants were asked to indicate the direction of the visual motion by pressing the corresponding (right or left) button. No response time limit was given. The stimulus presentation time started from 150 ms and was adjusted in the following trials using 'one-up two-down' staircases procedure with the initial step of 16.7 ms that decreased to 8.3 ms after the first two reversals. Separate staircases were used for the three types of the stimuli. The block continued until all staircases completed at least 7 reversals. The subject completed two blocks during each visit. The duration threshold was computed by averaging the presentation times over all the reversals excluding the first two, and then over the two blocks. For the purpose of the present study, only responses to the large (12˚) and small (1˚) stimuli were analyzed. As a result, for each visit and stimulus size we estimated the minimal exposure time required for the subject to discriminate the direction of motion of the small and large grating (in each visit: Threshold$_{SMALL}$ and Threshold$_{LARGE}$).

To estimate strength of perceptual suppression, we calculated the spatial suppression index (SSI) as:

$SSI$ = log10(Threshold$_{LARGE}$)–log10(Threshold$_{SMALL}$).

For a brief review of all the steps of psychophysical data analysis, see Fig 1 (lower panel).

## Statistical analyses

To analyze group and phase-related differences in gamma parameters, we used General Linear Models (GLM) implemented in the Rstatix and Stats packages in R 4.0.3 [100]. GR power values were log10-transformed to normalize the distributions. According to the Shapiro-Wilk test, distributions of the GR frequency and GR log10-transformed power did not differ significantly from normal (all p's>0.05). Since the preliminary analysis showed that the GR power and frequency depend on age (see below), standardized Age was included as a factor in the linear models for GR frequency and log10-transformed GR power. The between-group factors were Group (control, PMDD), Visit-Order (1$^{st}$ visit during the follicular phase, 1$^{st}$ visit during the luteal phase). The repeated measures factors were Phase (follicular, luteal) and Velocity (4 levels). We tested for the effects of Group, Visit-Order, Group x Visit-Order, Age, and their interactions with repeated measures factors Phase and Velocity. When appropriate, the Greenhouse-Geisser correction was used to adjust for the lack of sphericity. Planned comparisons were used to analyze the origin of significant repeated measures Analysis of Variance (ANOVA) effects.

To quantify changes in GR power from 'static' to 'slow' and from 'slow' to 'medium' conditions (see Results to substantiate the use of these contrasts), we also calculated 'GR facilitation' and 'GR suppression' as ratios:

$GR\ suppression$ = (1—GR Power$_{Medium}$/GR Power$_{Slow}$)*100%

$GR\ facilitation$ = (1—GR Power$_{Static}$/GR Power$_{Slow}$)*100%

A ratio metric can exaggerate between-group difference, especially if groups differ in initial value [101], which in this case corresponds to GR power in the 'slow' condition. Besides, a ratio yields reliable results only if the relationship between numerator and denominator is a straight line through the origin for at least one of the two groups being compared [101, 102].

These two requirements–no statistically significant difference in initial value and a zero inter-
cept–were fulfilled for both GR indexes (see S1 Fig for the detailed group comparison of GR
power). Therefore, the use of *GR suppression* and *GR facilitation* ratio metrics was statistically
justified.

Other steps of statistical analysis were performed using the standard Python 3.8.5 libraries
Numpy [103] and Scipy [104]. The data were tested for normality using Shapiro-Wilk test,
homogeneity of variance was evaluated using Levene's test. When assumptions for parametric
testing were violated, non-parametric tests were implemented, i.e., Wilcoxon signed-rank test
for within- and Mann-Whitney U test for between-samples comparisons. Alternatively,
parametric Student's t-test for related or independent groups was used.

Pearson correlation coefficient was estimated to assess the relationship between the nor-
mally distributed variables. When assumptions of normality were violated, Spearman's rank
correlation coefficient was estimated. Partial Pearson's and Spearman's rank correlation coeffi-
cients were used wherever necessary to control for confounding variables.

The Benjamini-Hochberg false discovery rate (FDR) correction with threshold 0.05 was
applied to the p-values to control for multiple tests.

## Results

### Premenstrual symptom severity in women with PMDD

Fig 2 shows the daily symptom severity scores in women with PMDD, averaged over two or
three MCs. For the subjects to meet criteria for PMDD on the C-PASS, at least five out of 20
symptoms need to meet the PMDD criteria, i.e., the symptoms need to be strong enough and
be reliably associated with the MC. The curves in Fig 2 represent averages over all 20

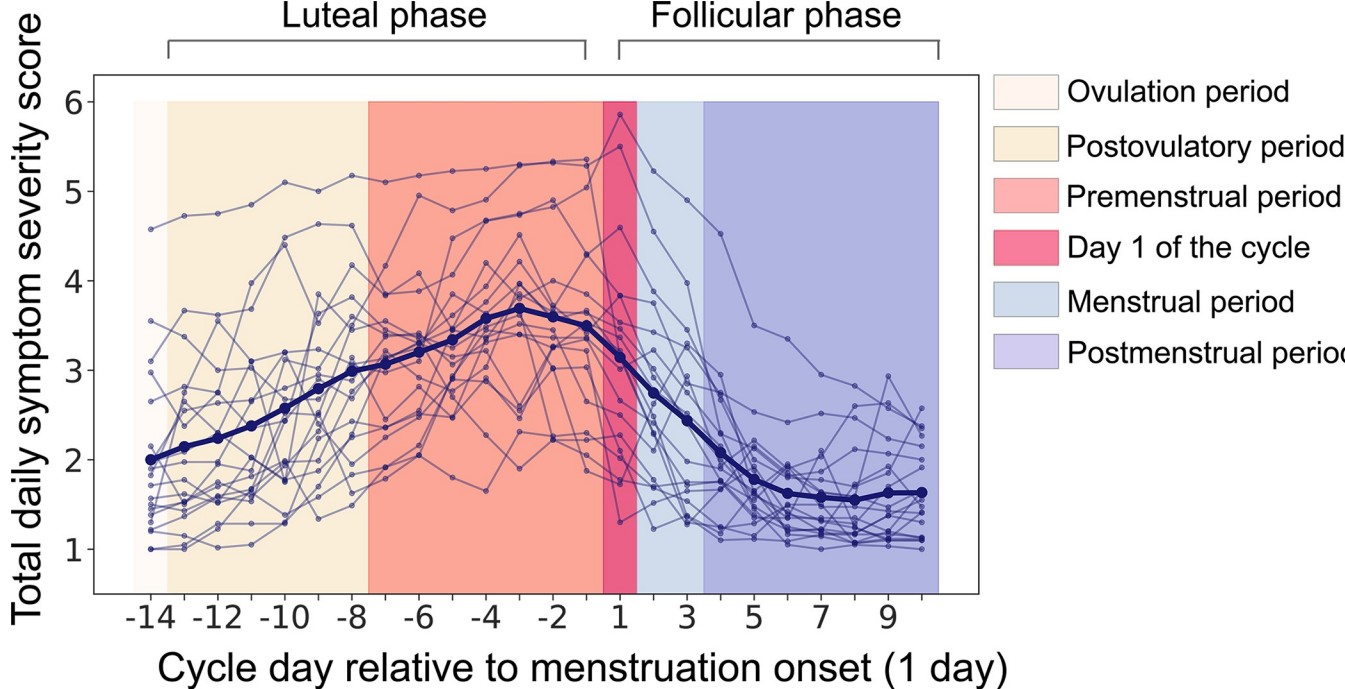

**Fig 2. Daily symptom severity scores (from 1 = 'Not at all' to 6 = 'Extreme') averaged over 20 symptoms (C-PASS questionnaire [88]) and two or three menstrual cycles in participants with PMDD.** The thin lines show individual participants; the thick line is the group average. The days are numbered relative to the start of menstruation (day 1).

**Table 2. State-Trait Anxiety Inventory (STAI) results in PMDD and control groups.**

| Scales | PMDD group Mean (S.D.) [range] | Control group Mean (S.D.) [range] | Group difference | |
| --- | --- | --- | --- | --- |
| | | | Mann-Whitney's U | *P*-value |
| STAI, trait $N_{PMDD}/N_{Control}$ = 20/27 | 47.23 (10.04) [30–72] | 39.83 (9.37) [27–62] | 163.5 | 0.011 |
| STAI, state (follicular) $N_{PMDD}/N_{Control}$ = 20/26 | 37.85 (8.09) [25–53] | 34.92 (10.83) [22–62] | 234.0 | 0.22 |
| STAI, state (luteal) $N_{PMDD}/N_{Control}$ = 20/27 | 47.6 (11.56) [26–71] | 34.33 (9.46) [20–56] | 105.5 | 0.0002 |

N–number of subjects; S.D.–standard deviation.

symptoms. It can be seen that symptom severity substantially increased during the week preceding the onset of menstruation (luteal phase; the score > = 3 of maximum 6) and then gradually decreased during the early follicular phase. According to a recent study of the PMDD subtypes [105], these average symptom scores indicate severe rather than mild PMDD in most participants in our clinical group.

**Psychometric characteristics of women with PMDD.** According to the BSDS (but not MDQ) questionnaire, 11 of our 20 participants with PMDD tested positive for bipolar disorder. According to the HCL-32 questionnaire, 13 of 20 participants with PMDD demonstrated hypomanic features. According to the BDI questionnaire, 6 out of 20 women with PMDD scored positive for mild-to-moderate depression in the follicular phase, whereas the frequency and severity of the depressive symptoms in the PMDD group strongly increased in the luteal phase. For more information on bipolar disorder and depression questionnaires, see S1 Table and S1 Text in S2 File.

According to the STAI questionnaire, the 'trait anxiety' was higher in women with PMDD than in control women (Mann-Whitney U test, p = 0.011, Table 2). The 'state anxiety' was higher in the PMDD than control group during the luteal phase (Mann-Whitney U test, p = 0.0002, Table 2), while no group differences were found during the follicular phase (Mann-Whitney U test, p = 0.22, Table 2). Also, women with PMDD had higher state anxiety when they visited the laboratory during the luteal compared to the follicular phase (Wilcoxon signed-rank test, Z = 40.0, p = 0.014), which was not the case in control women.

We expected our participants to be more anxious during their first visit to the laboratory, irrespective of the MC phase. Indeed, in the combined group of subjects, state anxiety was higher during the first visit compared to the second one (Mann-Whitney U = 319.5, p = 0.025). In the individual groups, however, the difference in the state anxiety between the first and second visits did not reach the level of significance (both p's>0.12).

## Plasma levels of steroids and PMDD symptom severity

In all participants, plasma level of progesterone increased from the follicular to the luteal phase (Fig 3A and 3B). Estradiol plasma level also increased from the follicular to the luteal phase in the majority of participants (Fig 3C and 3D). Thus, the hormone levels were in the range consistent with the follicular and luteal phases at each visit.

The luteal progesterone plasma level was reduced in women with PMDD (Student's t-test, t (45) = -2.53, p = 0.015; Fig 3F), while their luteal estradiol did not differ from that in the control group (Student's t-test, t(45) = 0.65, p = 0.52; Fig 3E). The ratio of estradiol to progesterone concentrations in the luteal phase was significantly elevated in the PMDD group (Mann-Whitney U = 150.0, p = 0.005; Fig 3G).

To test for the link between steroid hormone levels and symptom severity in PMDD, we estimated Pearson correlation between the hormone concentrations and the same-day PMS scores (number of days prior menstruation onset was partial out of the correlation). The

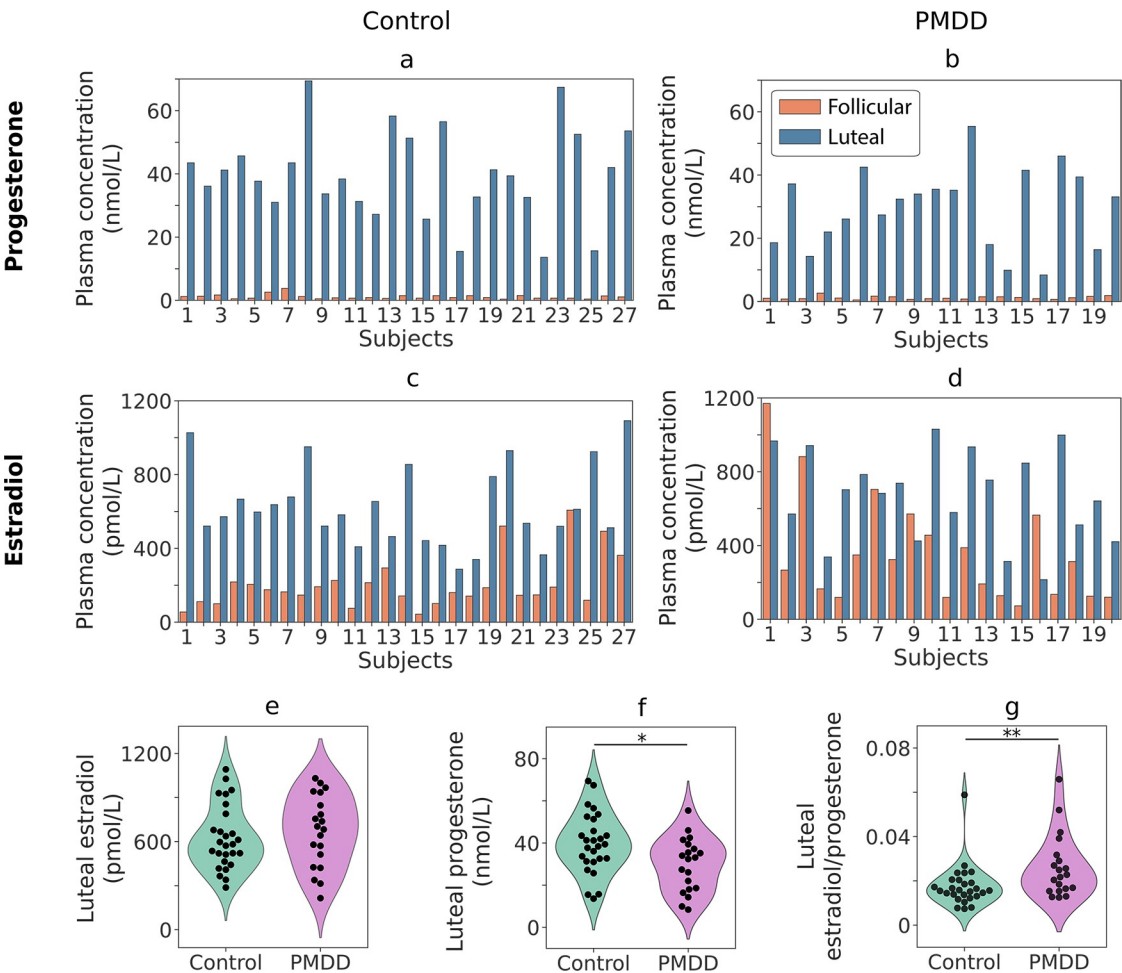

**Fig 3.** (a-d) Plasma levels of estradiol and progesterone in the control (a, c) and PMDD (b, d) groups in two phases of the menstrual cycle (follicular, luteal). (e-g) Violin plots show estradiol and progesterone concentrations and estradiol to progesterone ratios in the luteal phase of the menstrual cycle in control subjects and subjects with PMDD. $^*$p<0.05; $^{**}$p<0.01.

correlation was not significant ($N_{PMDD}$ = 18; estradiol: $r_{partial}$ = 0.32, p = 0.22; progesterone: $r_{partial}$ = 0.25, p = 0.33).

## MEG results

**Behavioral performance in MEG task.** The percentage of epochs with omission (reaction time<150 ms) and commission (no response or reaction time> = 1000 ms) errors did not differ significantly between the groups and MC phases (all p's>0.19). The reaction time was significantly longer in women with PMDD than in control participants during the luteal phase of the MC (Mann-Whitney U = 1995, p = 0.035), but not during the follicular phase (Mann-Whitney U = 2252.0, p = 0.23).

**Dependence of the GR parameters on age.** In line with the previous studies [106–108], GR frequency, averaged over phases, was negatively correlated with age in control women (Table 3). In women with PMDD, none of the correlations reached the level of significance. However, there was no significant difference in the correlation coefficients between the groups (all p's>0.15, two-sided). Phase-averaged GR power increased with age in both groups of participants and in all velocity conditions (Table 3).

**Table 3. Pearson's correlations between age and gamma response (GR) power (log10-transformed) and frequency in the two groups of participants (control and PMDD).** Gamma parameters were averaged over the two phases of the menstrual cycle.

| Grating's motion velocity | Control group (N = 27) | PMDD group (N = 20) |
|---|---|---|
| | *GR power* | |
| Static, 0˚/s | r = 0.62, p = 0.005 | r = 0.40, p = 0.11 |
| Slow, 1.2˚/s | r = 0.56, p = 0.008 | r = 0.46, p = 0.08 |
| Medium, 3.6˚/s | r = 0.65, p = 0.005 | r = 0.49, p = 0.07 |
| Fast, 6.0˚/s | r = 0.66, p = 0.005 | r = 0.54, p = 0.03 |
| | *GR frequency* | |
| Static, 0˚/s | r = -0.33, p = 0.13 | r = -0.06, p = 0.80 |
| Slow, 1.2˚/s | r = -0.38, p = 0.09 | r = -0.29, p = 0.26 |
| Medium, 3.6˚/s | r = -0.47, p = 0.03 | r = -0.23, p = 0.38 |
| Fast, 6.0˚/s | r = -0.34, p = 0.11 | r = 0.07, p = 0.80 |

N—number of subjects; p-values are FDR-corrected for multiple comparisons.

**Effects of diagnosis, MC phase, and visit order on GR frequency.** Results of GLM with factors Group (control, PMDD), Visit-Order (1st visit during the follicular phase, 1st visit during the luteal phase), repeated measures factors Phase (follicular, luteal) and Velocity (4 levels), and Age as a covariate are presented in Tables 4 and 5 for GR frequency and GR power, respectively. S1 Fig shows violin plots of individual GR frequency values for each group, condition, and MC phase.

There was a highly significant effect of *Velocity*: GR frequency almost linearly increased with increasing grating's motion velocity (Table 4, Fig 4A and 4B). There was also a significant effect of *Phase* (luteal>follicular), which however can be better understood by inspecting the highly significant interaction between *Phase* and *Visit-Order* (F(1,42) = 18.0, p = 0.0001). This result shows that the difference in GR frequency between the luteal and follicular phases was

**Table 4. ANCOVA results for GR frequency.**

| Effect | F | df | G-G epsilon | Generalized eta squared | P / adj. p |
|---|---|---|---|---|---|
| Group | 0.159 | 1, 42 | | 0.003 | 0.69 |
| Visit-Order | 0.142 | 1, 42 | | 0.003 | 0.71 |
| Group * Visit-Order | 0.520 | 1, 42 | | 0.009 | 0.47 |
| **Phase** | **8.259** | **1, 42** | | **0.004** | **0.006** |
| Phase * Group | 0.000 | 1, 42 | | $3.7^*10^{-8}$ | 0.99 |
| **Phase * Visit-Order** | **17.968** | **1, 42** | | **0.009** | **0.0001** |
| Phase * Group * Visit-Order | 0.105 | 1, 42 | | 0.00005 | 0.75 |
| **Velocity** | **410.088** | **3, 126** | **0.61** | **0.63** | **$2.5^*10^{-40}$** |
| Velocity * Group | 0.185 | 3, 126 | 0.61 | 0.0008 | 0.81 |
| Velocity * Visit-Order | 0.859 | 3, 126 | 0.61 | 0.004 | 0.42 |
| Velocity * Group * Visit-Order | 0.747 | 3, 126 | 0.61 | 0.003 | 0.47 |
| Phase * Velocity | 0.347 | 3, 126 | 0.58 | 0.0003 | 0.68 |
| Phase * Velocity * Group | 0.078 | 3, 126 | 0.58 | 0.00006 | 0.90 |
| Phase * Velocity * Visit-Order | 2.845 | 3, 126 | 0.58 | 0.002 | 0.072 |
| Phase * Velocity * Group * Visit-Order | 0.248 | 3, 126 | 0.58 | 0.0002 | 0.75 |

Significant effects are highlighted in bold.

**Table 5. ANCOVA results for log10-transformed GR power.**

| Effect | F | df | G-G epsilon | Generalized eta squared | P / adj. p |
|---|---|---|---|---|---|
| Group | 0.833 | 1, 42 | | 0.014 | 0.37 |
| Visit-Order | 0.608 | 1, 42 | | 0.010 | 0.44 |
| Group * Visit-Order | 3.502 | 1, 42 | | 0.055 | 0.068 |
| Phase | 0.070 | 1, 42 | | 0.0002 | 0.79 |
| Phase * Group | 0.155 | 1, 42 | | 0.0004 | 0.70 |
| Phase * Visit-Order | 0.909 | 1, 42 | | 0.002 | 0.35 |
| Phase * Group * Visit-Order | 2.714 | 1, 42 | | 0.007 | 0.11 |
| **Velocity** | **152.058** | **3, 126** | **0.71** | **0.375** | **$3.5^*10^{-30}$** |
| **Velocity * Group** | **3.464** | **3, 126** | **0.71** | **0.013** | **0.033** |
| Velocity * Visit-Order | 1.661 | 3, 126 | 0.71 | 0.007 | 0.19 |
| Velocity * Group * Visit-Order | 0.618 | 3, 126 | 0.71 | 0.007 | 0.55 |
| Phase * Velocity | 1.615 | 3, 126 | 0.76 | 0.0008 | 0.20 |
| Phase * Velocity * Group | 2.441 | 3, 126 | 0.76 | 0.001 | 0.085 |
| **Phase * Velocity * Visit-Order** | **3.020** | **3, 126** | **0.76** | **0.001** | **0.047** |
| Phase * Velocity * Group * Visit-Order | 0.317 | 3, 126 | 0.76 | 0.0002 | 0.76 |

Significant effects are highlighted in bold.

present only if the luteal phase occurred during the first visit to the MEG laboratory (Fig 4C and 4D, right panels), but not during the second visit (Fig 4C and 4D, left panels). Notably, the effect of *Phase* and *Visit-Order* interaction–higher GR frequency during the luteal phase if it coincided with the first visit–was the same in both PMDD and control groups (Fig 4C and 4D). Indeed, the *Phase* and *Visit-Order* interaction remained highly significant when tested separately in the control and PMDD groups (Control: F(1,24) = 16.3, p = 0.0005; PMDD: F(1,17) = 11.5, p = 0.003). The effect of *Group* and interactions of *Group* with *Phase*, *Visit-Order*, or *Velocity* were all not significant.

**Effects of diagnosis and MC phase on GR power.** The group average GR power spectra for participants from the control and PMDD groups are shown in Fig 5. S1 Fig shows violin plots of individual GR power values for each group, condition, and MC phase.

For GR power, there was a highly significant effect of *Velocity* (Table 5): the power increased from the 'static' to the 'slow' condition and then decreased with further increase of grating's drift rate (Fig 5). There were no main effects of *Phase* or *Group*. However, there was a significant *Velocity* and *Group* interaction (F(3, 126) = 3.5, epsilon = 0.71, adj. p = 0.033). There was also a tendency for *Phase* * *Velocity* * *Group* interaction (F(3, 126) = 2.4, epsilon = 0.76, adj. p = 0.085). Since we had a clear prediction of a reduction of GR suppression in women with PMDD during the luteal phase and of the association between this reduction and PMS severity, we further analyzed this prediction even though the *Phase* * *Velocity* * *Group* interaction was only a trend.

**Modulations of GR power by visual motion velocity.** To investigate group differences in velocity-related modulations of GR power, we used post-hoc planned comparisons. There were no group differences in GR powers in any MC phase or velocity condition (all p's>0.1). However, suppression of the GR power from the 'slow' to the 'medium' condition was lower in women with PMDD than in control women during the luteal phase (F(1, 42) = 5.1, p = 0.03; Fig 6A, right panel). This group difference was absent during the follicular phase (F(1, 42) = 0.01, p = 0.9; Fig 6A, left panel). Fig 6C shows this *GR suppression* in percent of GR power in the 'slow' condition, which induced maximal power of GR at the group level; *GR suppression* =

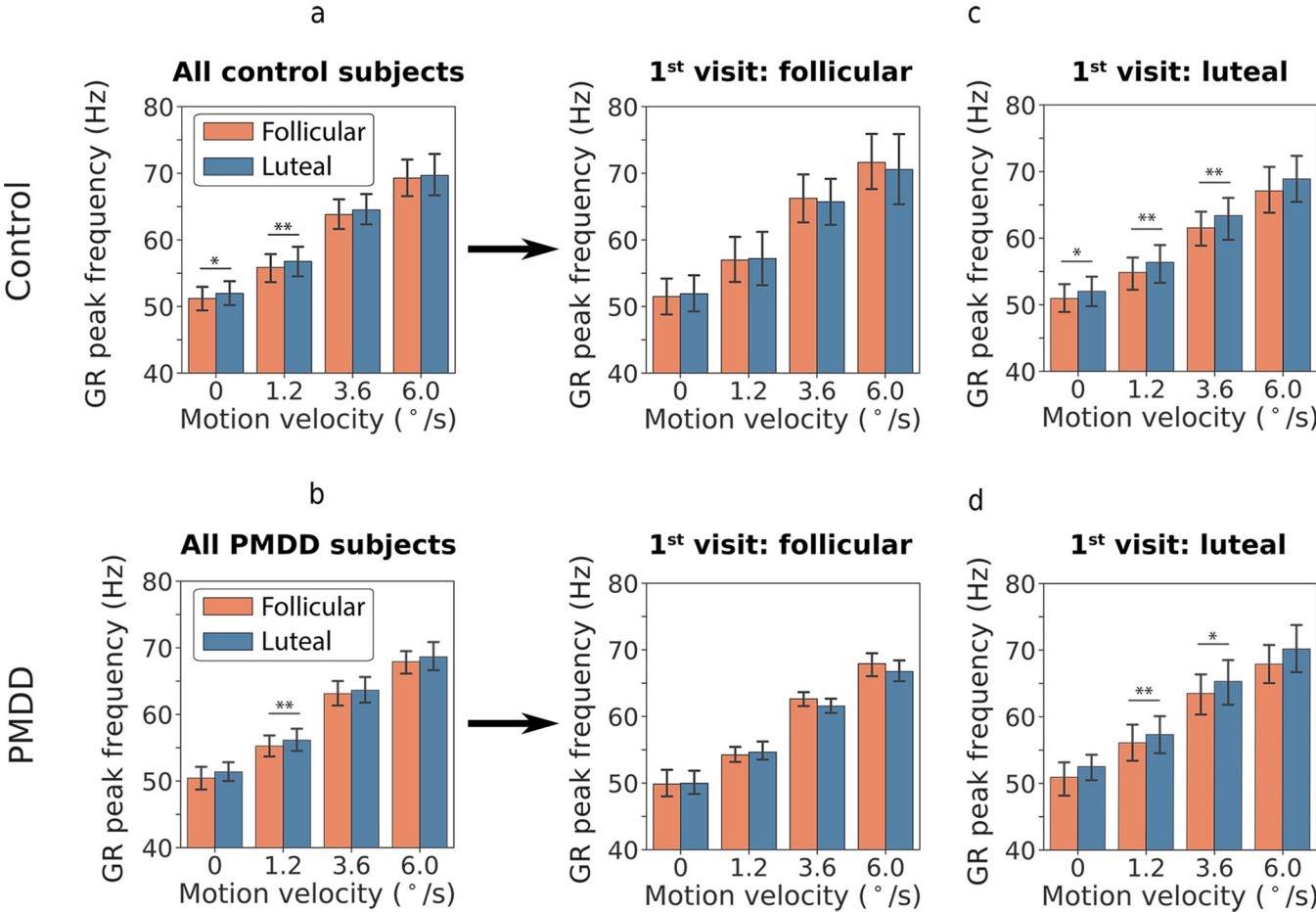

**Fig 4. Effect of the phase of the menstrual cycle on the weighted peak frequency of gamma response (GR).** The data are shown for four motion velocity conditions ('static': 0°/s, 'slow': 1.2°/s, 'medium': 3.6°/s, 'fast': 6.0°/s). (a) and (b) show comparison of the GR weighted peak frequency in the luteal and follicular phases separately for all control (a) and all PMDD (b) subjects. (c, d) show comparison of the GR weighted peak frequency in the luteal and follicular phases separately for those subjects, who came for the first investigation during their follicular (c, d left panel) or luteal (c, d right panel) phase. Error bars represent 95% confidence intervals. *p<0.05; **p<0.01.

(1—GR Power$_{Medium}$/GR Power$_{Slow}$)*100%. Larger values indicate stronger GR power decrease under higher sensory load. S2 Fig shows violin plots of individual GR suppression scores for each group and MC phase.

The GR suppression increased from the follicular to the luteal phase in control women (Student's t-test, t(26) = 2.2, p = 0.041; Fig 6C), but not in women with PMDD (Student's t-test, t(19) = -1.5, p = 0.16; Fig 6C), in which the direction of change was even inverted. As a result, in the luteal phase women with PMDD had lower GR suppression than control participants (Student's t-test, t(45) = -2.5, p = 0.017; Fig 6C). The lower luteal GR suppression in women with PMDD correlated with higher severity of their same-day PMS scores (N$_{PMDD}$ = 18, Pearson's r = -0.49, p = 0.04; Fig 6D).

We then tested for the presence of group differences in *GR facilitation*, i.e., an increase of GR power from the 'static' to the 'slow' condition, which likely has a different mechanism from that of GR suppression (see [65] and Discussion section). The increase of GR power from the 'static' to the 'slow' condition was steeper in PMDD than in control participants in both MC phases (follicular: F(1, 43) = 4.2, p = 0.047; luteal: F(1, 43) = 7.0, p = 0.01; combined phases: F(1, 43) = 6.1, p = 0.02; Fig 6A). This means that women with PMDD show a stronger

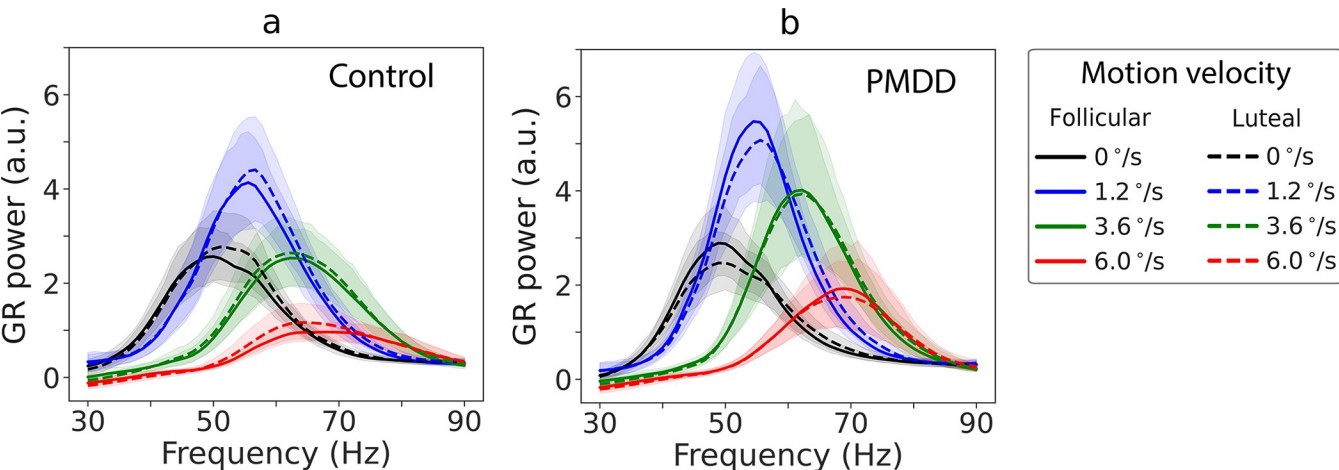

**Fig 5. Grand average spectra of gamma response (GR) ([stimulation-prestimulus] / prestimulus).** The spectra are shown for four motion velocity conditions ('static': 0˚/s, 'slow': 1.2˚/s, 'medium': 3.6˚/s, 'fast': 6.0˚/s), two phases of the menstrual cycle (follicular, luteal), and two groups of subjects (control (a), PMDD (b)). Shaded areas represent 95% confidence intervals.

facilitation of GR power from the 'static' to the 'slow' condition regardless of the MC phase. Fig 6B presents GR facilitation in percent of GR power in the 'slow' condition, which induced maximal power of GR at the group level: *GR facilitation* = (1—GR Power$_{Static}$/GR Power$_{Slow}$)* 100%. Larger values indicate stronger GR facilitation with a moderate increase of sensory load. S2 Fig shows violin plots of individual GR facilitation scores for each group and MC phase. Similarly to the planned comparison analysis, this measure indicates greater GR facilitation in PMDD than in control subjects during both follicular (Student's t-test, $t(45) = 2.2$, $p = 0.034$) and luteal (Student's t-test, $t(45) = 2.9$, $p = 0.006$) phases of the MC (Fig 6B). The magnitude of GR facilitation did not change between MC phases in either the control (Student's paired t-test, $t(52) = 0.16$, $p = 0.87$) or PMDD group (Student's paired t-test, $t(38) = 0.62$, $p = 0.54$).

To sum up, we found that the modulation of GR power by velocity differed in women with PMDD and control participants. GR facilitation from the 'static' to the 'slow' condition (i.e., on the ascending branch of the GR power modulation curve) was greater in women with PMDD than in control women during both phases of the MC. On the other hand, GR suppression from the 'slow' to the 'medium' velocity condition (i.e., at the descending branch of the GR power modulation curve) was reduced in PMDD participants specifically during the luteal phase, and correlated with the severity of PMS on the day of the visit.

**Relationship between GR power and GR frequency modulations.** We have previously reported a significant correlation between an increase in GR frequency and suppression of GR power from the 'slow' to the 'medium' visual motion velocity condition [64].

Consistently with the previous findings, we found in control participants a correlation between the 'slow'/'medium' GR power ratio and an increase in GR frequency from the 'slow' to the 'medium' condition (N$_{Control}$ = 27, follicular: Pearson's r = 0.39, p = 0.04; luteal: Pearson's r = 0.56, p = 0.002). This correlation was absent in the PMDD sample (N$_{PMDD}$ = 20, follicular: Pearson's r = -0.15, p = 0.5; luteal: Pearson's r = -0.08, p = 0.7). The between-group difference in correlation coefficients was significant in the luteal phase (p = 0.03) and approached significance level in the follicular phase (p = 0.09).

**Relationship between GR parameters and plasma levels of steroid hormones.** To test for the link between plasma levels of steroid hormones and GR parameters, we estimated partial Spearman's correlation coefficients between GR frequency or GR power in the

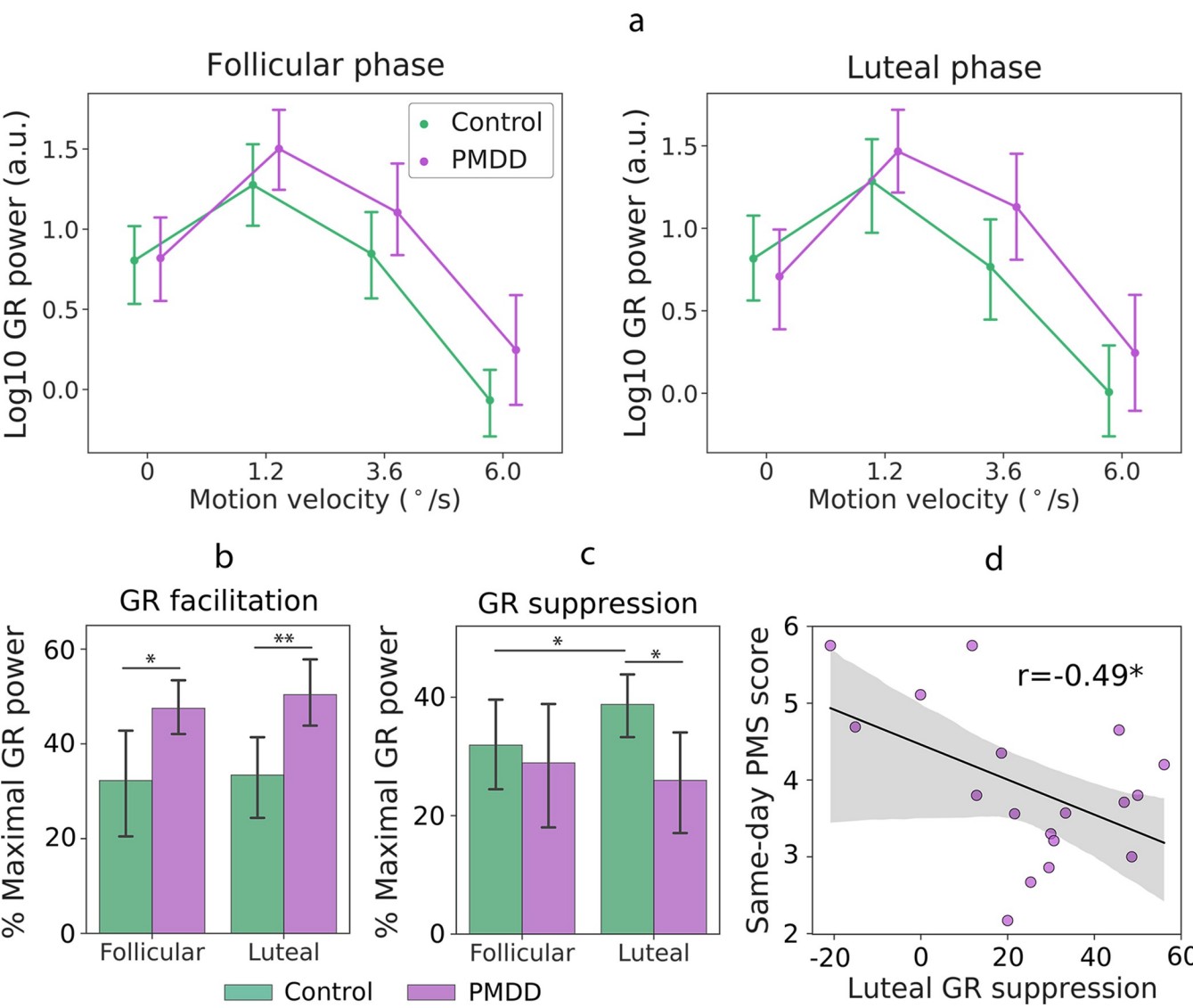

**Fig 6. Group differences in gamma response (GR) power.** (a) Log10-transformed GR power modulation in four velocity conditions ('static': 0°/s, 'slow': 1.2°/ s, 'medium': 3.6°/s, 'fast': 6.0°/s) in the follicular (left panel) and luteal (right panel) phases of the menstrual cycle. (b) Group differences in velocity-related GR power facilitation from the 'static' to the 'slow' velocity condition, estimated in percent of the GR power in the 'slow' condition, which induced maximal GR at the group level: (1—GR Power$_{Static}$/GR Power$_{Slow}$)*100%. (c) Group differences in velocity-related GR power suppression from the 'slow' to the 'medium' velocity condition, estimated in percent of the GR power in the 'slow' condition: (1—GR Power$_{Medium}$/GR Power$_{Slow}$)*100%. (d) Correlation between luteal GR power suppression and premenstrual symptom (PMS) severity measured on the same day in women with PMDD. Error bars and shaded areas represent 95% confidence intervals.

corresponding phase of the MC and (A) follicular estradiol, (B) luteal estradiol, or (C) luteal progesterone separately in the control and PMDD groups (see S1 Table). Age was partialled out of the correlations because it affected the GR parameters (see Table 3). Correlations with follicular progesterone were not assessed because the poor sensitivity of commercial immuno-assay systems to low progesterone concentrations (<5 nmol/L) precludes its reliable measure-ment [109]. In control participants, there were positive correlations between follicular estradiol and GR frequency in the 'medium' and 'fast' velocity conditions. None of these corre-lations, however, survived correction for multiple comparisons.

Results of the psychophysical experiment: Directional sensitivity to motion.

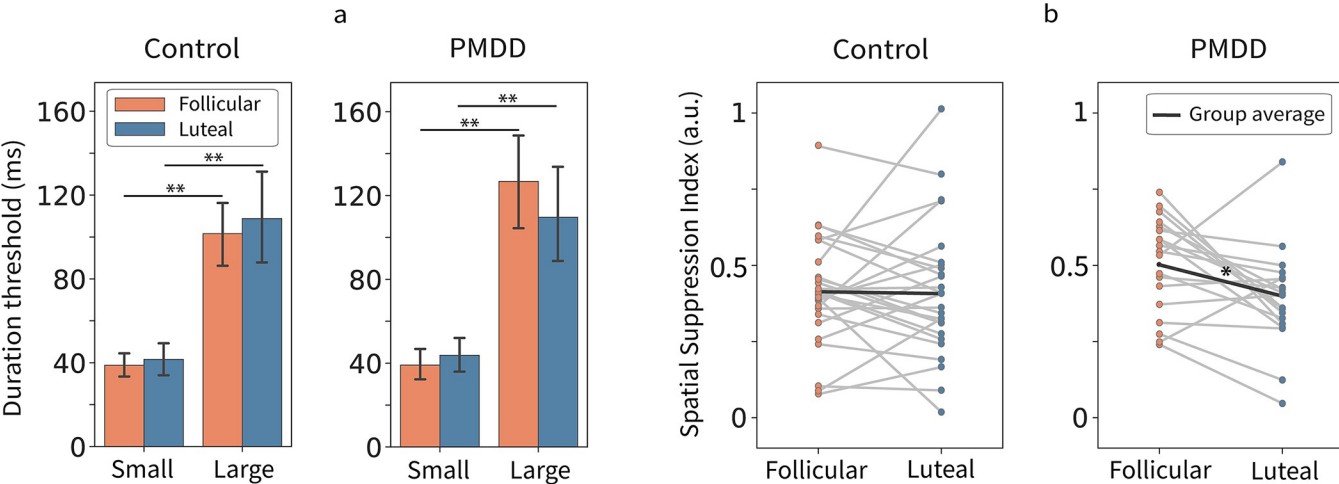

**Fig 7. Dependence of motion direction distinction on the phase of the menstrual cycle (follicular, luteal phase) in control and PMDD participants.** (a) duration thresholds for the small and large stimuli (b) and spatial suppression index (SSI). Error bars represent 95% confidence intervals. *p<0.05; **p<10⁻⁸.

Spatial suppression was estimated in 19 of 20 subjects with PMDD and in 26 of 27 control subjects. Two women (1 control, in both phases; 1 PMDD, in the follicular phase) demonstrated a persistent illusion of reverse motion when presented with large moving grating, which did not allow to reliably estimate motion direction discrimination thresholds (hereafter, 'duration thresholds'), in these participants.

In both groups and during both phases of the MC, subjects demonstrated reliable perceptual spatial suppression: it took them longer to discriminate direction of motion of a large than of a small visual grating (all F's>9.0, p's<0.0001; Fig 7A). Based on duration thresholds, we calculated the spatial suppression index (SSI; SSI = log10(Threshold$_{LARGE}$/Threshold$_{SMALL}$)) and tested for the *Group* and *Phase* differences in spatial suppression using repeated measures ANOVA. This analysis revealed the main effect of *Phase* ($F(1,43) = 4.37$, $p = 0.043$) and *Group* * *Phase* interaction ($F(1,43) = 4.10$, $p = 0.049$), which is illustrated in Fig 7B. In the PMDD group, SSI decreased from the follicular to the luteal phase (Student's t-test, $t(18) = 2.49$, $p = 0.022$). Importantly, this decrease was mainly due to an *improvement* in discriminating the direction of motion for the *large* stimulus during the luteal phase (mean follicular Threshold$_{LARGE}$ = 127 ms, mean luteal Threshold$_{LARGE}$ = 110 ms; Student's t-test, $t(18) = 2.02$, $p = 0.06$), rather than a change in the duration threshold for the small one (mean follicular Threshold$_{SMALL}$ = 39 ms, mean luteal Threshold$_{SMALL}$ = 44 ms; Student's t-test, $t(18) = 1.3$, $p = 0.19$). There was also a tendency for stronger spatial suppression in women with PMDD than in control women during the follicular phase (Student's t-test, $t(18) = 1.8$, $p = 0.08$; Fig 7B).

## Relationship between perceptual spatial suppression and GR power suppression

In control women, the percent of GR suppression from the 'slow' to the 'medium' condition (Fig 6C) correlated with the SSI during the follicular phase (N$_{Control}$ = 26; Pearson's $r = 0.53$, $p = 0.005$). During the luteal phase, the correlation was in the same direction, although not significant (N$_{Control}$ = 26; Pearson's $r = 0.31$, $p = 0.12$). The difference between correlation coefficients in the follicular and luteal phases in the control group was not significant. This direction of correlations is in line with results of our previous study in a smaller sample of neurotypical women (13 of them were included in the control sample in this study) where we estimated GR

suppression as a 'gamma suppression slope' based on three velocity conditions [63]; for the same analysis of the present data see the S3 Table and S2 Text in S3 File). In the PMDD group, this correlation was absent ($N_{PMDD}$ = 19; follicular Pearson's r = -0.12, p = 0.6; luteal Pearson's r = -0.25, p = 0.3). The group difference in correlation coefficients was significant in the follicular phase (p = 0.035, two-sided) but not in the luteal phase (p = 0.084, two-sided).

## Discussion

We hypothesized that in women with PMDD, the paradoxical response to neurosteroids, which results in changes of neural E/I balance during the luteal phase of the MC, would affect parameters of visual gamma oscillations and visual perception that are highly dependent on E/I balance, and that these abnormalities would be specific to the symptomatic luteal phase. To test this hypothesis, in women with and without PMDD we measured visual gamma responses (GRs) to drifting gratings with MEG and, in a separate psychophysical experiment, estimated perceptual spatial suppression. By using drift rate of a visual grating as a proxy of excitatory drive and changing it from 'static' to 'slow' and further to 'medium' and 'fast' [65], we probed the dynamics of gamma response, which strongly depends on the neural E/I balance.

Our predictions were only partially confirmed. The only parameter that distinguished women with PMDD from women with the healthy MC specifically during the luteal phase was the attenuated suppression of GR from the 'slow' condition optimal for GR generation to the 'medium' one. This attenuation indicates an abnormally decreased inhibition during the luteal phase in women with PMDD compared to control women. Reduced GR suppression during the luteal phase in women with PMDD correlated with greater severity of their clinical symptoms assessed on the same day. Another characteristic feature of the PMDD group was an atypically strong facilitation of GR power from the 'suboptimal' for gamma generation 'static' to the 'optimal' 'slow' condition. This excessive GR facilitation, however, was observed in both MC phases, indicating a constitutively elevated E/I ratio in the visual cortex in women with PMDD regardless of the effect of neurosteroid hormones.

Contrary to our expectations, GR frequency was not affected in women with PMDD. Their perceptual spatial suppression was not altered in the expected direction (i.e., was not decreased). On the contrary, in the follicular phase, spatial suppression in women with PMDD tended to be stronger than in control women, but it significantly decreased from the follicular to the luteal phase. In the discussion that follows, we will argue that in women with PMDD, neural excitability in the visual cortex is constitutively elevated and that the E/I balance shifts even more toward excitability during the luteal phase, possibly due to tolerance to ALLO.

### Representativeness of the PMDD sample

It has been suggested that the often inconsistent findings in PMDD may be due, at least in part, to inconsistent diagnostic practices and varying symptom severity in patients [88, 110, 111]. We therefore start the discussion with the characterization of our PMDD sample.

To confirm the presence of PMDD and assess its severity, we used a questionnaire in which participants reported their symptoms daily for two or three MCs. This approach provides a more reliable assessment of PMDD than questionnaires based on retrospective PMS reports [88, 112]. Consistent with the previously described PMDD symptomatology [88, 113, 114], the severity of self-reported symptoms in participants of the clinical group increased from the ovulatory to the late luteal phase and decreased after the onset of menstruation (Fig 2). In a similar vein, according to the STAI and BDI questionnaires, their state anxiety and depressive symptoms, which are diagnostic features of PMDD [89], also exacerbated during the luteal phase. The state anxiety scores of women with PMDD were higher than those of control participants

specifically during the luteal phase (Table 2). In addition to MC-dependent mood changes, high scores on trait anxiety, depression, and bipolar disorder questionnaires (according to their cut-off scores) in the PMDD sample agree well with previous reports of a high frequency of respective comorbid disorders in this clinical group [86].

A comparison of our participants' daily PMS scores with those in the Eisenlohr-Moul et al's study suggests that the majority of participants in our clinical sample had severe rather than mild form of PMDD (compare Fig 2 in the present paper and Fig 1 in [105]). Abnormal luteal hormone levels in women with PMDD compared with age-matched control participants are also consistent with the presence of severe PMDD in the clinical group. Indeed, there is evidence that decreased luteal progesterone, together with an elevated luteal estradiol/progesterone ratio in the mid-luteal phase of the MC (Fig 3F and 3G) characterize women with severe rather than mild PMS [111].

Thus, we discuss below the results obtained in a sample of women with severe PMDD and age-matched control women.

## Altered modulation of MEG visual GR power by excitatory drive in PMDD

**Phase-dependent changes in GR suppression in control women and in those with PMDD.**   In control women, changes in the visual GR power caused by the increase of excitatory drive were affected by the MC phase on the descending branch of the GR power modulation curve: during transition from the 'slow' to the 'medium' velocity (Fig 6C). The suppression of GR with an increase in excitatory drive from the 'slow' to the 'medium' and then to the 'fast' drift rate most likely reflects desynchronization of gamma oscillations as a result of compensatory 'excessive' activation of inhibitory neurons ([68, 115, 116]; see [65] for a detailed discussion). Indeed, animal studies have shown that increasing excitatory drive to the visual cortex above a certain limit leads to a greater increase in the inhibitory than excitatory current on the principal neurons, so the overall E/I ratio shifts toward inhibition [115]. Hence, a steeper slope of GR suppression at stimulation intensities exceeding the 'optimal' one for gamma generation may indicate a transition to a stronger inhibition of the principal neurons.

Despite the presence of phase-related changes in GR suppression in women with a healthy MC, their absolute GR power did not differ between the follicular to the luteal phases. The later result is consistent with that of Sumner et al. [55], and underscores the functional relevance of velocity-related *modulation* of power of gamma oscillations as a measure of the E/I ratio in the visual cortex. Considering the presumed role of strong inhibition in GR suppression, the increase in GR suppression in the luteal phase compared to the follicular phase of the MC in control women is consistent with enhanced inhibitory transmission. The increase in inhibition associated with the luteal phase may result from potentiation of the tonic GABA$_A$ conductance in principal neurons by ALLO, the neuroactive metabolite of progesterone. ALLO concentration follows that of progesterone [117] and is elevated during the luteal phase. ALLO targets the GABA$_A$Rs family [118], but especially strongly potentiates δGABA$_A$Rs (δGABA$_A$Rs) [19–21]. These receptors are located extrasynaptically and produce a tonic form of inhibition [20], which plays an important role in controlling neuronal excitability [119] and gamma generation [116, 120]. Furthermore, the elevated concentration of ALLO during the luteal phase affects GABA$_A$Rs expression pattern: it potentiates the expression of δGABA$_A$Rs and therefore further augments the effect of ALLO on tonic inhibitory neurotransmission [30, 121].

The putative enhancement of inhibitory neurotransmission from the follicular (i.e., low ALLO) to the luteal (i.e., high ALLO) phase of the MC is consistent with studies in rodents

that demonstrate a potentiating effect of physiological levels of ALLO on inhibitory neurotransmission during the estrous cycle [30, 121, 122]. Evidence in favor of increased inhibitory tone during high-ALLO mid-luteal phase also comes from TMS studies in the motor cortex of naturally cycling women [40, 123, 124] and from reports of decreased seizure susceptibility during this phase of the MC [13]. However, the effect of ALLO on net excitability may vary between brain regions and species [125–127]. As for the human visual cortex, the results by Epperson et al. [42] are particularly relevant to our finding. In a magnetic resonance spectroscopy study, these authors showed a significant decrease of GABA concentration in the occipital cortex from the follicular to the mid-luteal phase of the healthy MC and interpreted this result in terms of homeostatic regulation, which partially compensates for the increased sensitivity of GABA$_A$Rs to ALLO by depressing GABA synthesis.

Given the scarce and inconsistent literature on the role of neurosteroid hormones in the human visual cortex [43, 128], our results pointing to increased inhibitory tone in gamma-generating visual circuitry during the luteal phase in healthy MC are particularly important. They strengthen the evidence linking the physiological effect of ALLO with increased tonic inhibition in the visual cortex of healthy naturally cycling women.

In contrast to control participants, women with PMDD did not show strengthening of GR suppression from the follicular to the luteal phase of the MC (Fig 6C). The absence of this normal change in the GR suppression suggests the lack of normal enhancement of neural inhibition in the visual cortex in women with PMDD during their symptomatic luteal phase. This result resembles that obtained by Smith and colleagues in the motor cortex of women with strong PMS [40]. Using TMS, Smith et al. found in these women a luteal phase-specific deficit in cortical inhibition and explained it by the lack of the normal increase in inhibition during the luteal phase.

Since our results in women with PMDD suggest the lack of normal luteal increase in GR suppression (indicative of unchanged inhibition between the follicular and luteal phases) rather than luteal decrease in GR suppression (which would be indicative of reduced inhibition in the luteal phase), they are more compatible with developing tolerance to ALLO [17] than with its paradoxical excitatory effect [21]. This tolerance may reflect an abnormally attenuated action of ALLO on δGABA$_A$Rs on principal neurons, possibly due to an abnormal or insufficient plasticity of the δGABA$_A$Rs in response to the postovulatory increase in ALLO [17, 19]. Indeed, reduced sensitivity to the sedative effects of GABA$_A$Rs agonists, such as benzodiazepines, ethanol, and GABA-active steroids (pregnanolone), was previously demonstrated in women with PMDD during the luteal phase, when ALLO concentration is high, but not during the follicular phase, when the levels of GABA-active steroids are low [16, 129–131].

Although changes in visual circuitry are unlikely to be directly related to the core PMDD symptoms (i.e., depression, anxiety, irritability, and mood lability), the luteal phase-specific deficit in visual GR suppression in PMDD was associated with more severe clinical symptoms assessed on the same day (Fig 6D). This correlation can be explained by abnormal action of progesterone-derived neurosteroids, i.e., ALLO, on δGABA$_A$Rs in multiple brain structures, which affects gamma oscillations via altering inhibitory neurotransmission in the visual cortex and is manifested in clinical symptoms of PMDD via altering inhibitory neurotransmission in the brain regions involved in the control of mood and anxiety (see, e.g., [132–134]).

To sum up, our results indicate that the GR suppression associated with increasing excitatory drive to the visual cortex reflects a luteal phase-specific and symptom severity-related deficit in neural inhibition in PMDD.

**GR facilitation in control women and in those with PMDD.** The GR facilitation caused by a moderate increase in excitatory drive (i.e., from static to slowly drifting grating) most probably has a different mechanism than the GR suppression caused by its strong increase

(when drift rate of the grating changes from 'slow' to 'medium' or 'fast') (see [61, 65] for discussion). While GR suppression is thought to be associated with an 'excessive' activation of inhibitory neurons [68], GR facilitation reflects increased activation of the excitatory principal cells and their synchronization [57, 135]. When sensory input increases to a certain level, more excited cortical circuitries produce stronger gamma oscillations in both animals [136, 137] and humans [138, 139].

In contrast to GR suppression, GR facilitation showed no MC-related changes in either control women or women with PMDD. The dissociation between the GR facilitation and GR suppression in their sensitivity to cyclic changes in neurosteroids suggests a difference in their mechanisms. Yet, the enhanced GR facilitation differentiated women with PMDD from control participants in both phases of the MC. Although this result was not predicted, it suggests the existence of additional MC-independent mechanisms involved in PMDD.

Both principal cells and inhibitory interneurons play pivotal roles in generation of gamma [57, 135]. However, normal peak frequency of the GR in women with PMDD (Table 4) speaks against an apparent deficit in functioning of the inhibitory neurons (see later in this Discussion). Therefore, the atypically high GR facilitation in PMDD is rather due to an elevated excitability of the principal cells and their enhanced propensity to engage in synchronous oscillations in response to visual stimulation (i.e., 'increased gain'). A similar deficit in gain regulation was observed in the visual cortex of patients with idiopathic generalized epilepsy using the steady-state visually evoked potentials (SSVEP) paradigm. These patients demonstrated an abnormally strong increase of SSVEPs associated with increasing contrast of a flickering grating [140–142]. The authors concluded that the altered regulation of gain control in these patients is the result of hyperexcitability of local neuronal ensembles and enhanced lateral spread of the excitation.

Because GR facilitation did not change between the follicular and luteal phases in either control or PMDD women, but was elevated in PMDD women during both of these MC phases, it is unlikely to depend on neurosteroid changes in MC and may reflect constitutively elevated excitability of the principal neurons in women with PMDD.

Our finding of a neurofunctional abnormality that is not limited to the symptomatic luteal phase in women with PMDD is not unique. Several studies that applied a range of different methods and paradigms reported in women with PMDD neurofunctional abnormalities during asymptomatic period of the MC [16, 33, 37, 38, 42, 129–131, 143–145]. Significantly decreased GABA concentration in the visual cortex was found in women with PMDD during the follicular phase [42]. There is also evidence on altered activation in the dorsolateral prefrontal-cingulate cortex [38] and enhanced amygdala response to negative stimuli [145] in women with PMDD regardless of the MC phase. Thus, our results provide further support for the idea that pathophysiology of PMDD is not limited to the postovulatory luteal phase events, and that E/I imbalance in PMDD is present even during the asymptomatic follicular phase, but is further exacerbated during the premenstrual period, presumably due to altered sensitivity to GABA-active neurosteroids.

Because our study is limited to a relatively small sample of women with PMDD, some of whom may have had undiagnosed comorbid neuropsychiatric conditions (e.g., depression), it does not allow us to conclude whether their elevated GR facilitation, which was observed in both luteal and follicular phases, is specific for PMDD. Various neuropsychiatric disorders are thought to be associated with constitutively altered regulation of E/I balance in visual cortex [146, 147], which in turn can contribute to visual perceptual abnormalities frequently observed in these disorders [81, 148–152]. Further studies that include the control groups of patients with these conditions without past or present PMDD are therefore needed.

**Frequency of GR and first scan effect.** The neural mechanisms regulating peak frequency of gamma oscillations are substantially different from those regulating their power. While power of gamma oscillations reflects synchronized activity of pyramidal cells, their frequency is predominantly controlled by tonic excitability of parvalbumin-containing (PV+) inhibitory neurons [66, 70, 153, 154]. Difference in the mechanisms that regulate GR frequency and power may explain their different dynamics associated with increasing excitatory drive [64, 136, 137].

Similarly to Sumner et al. [55], we found that in women with healthy MC, frequency of GR induced by static and slowly moving gratings increased from the follicular to the luteal phase (Fig 4A). However, a closer investigation showed that this effect was exclusively driven by those subjects who first came for the MEG experiment during the luteal phase (Fig 4C, right) and was absent in those who first came during the follicular phase (Fig 4C, left). Although Sumner et al. counterbalanced phases of the first visit (luteal 1st vs. follicular 1st), they did not test the effect of the order of visits, which, in our study, appears to be an important factor associated with the luteal phase-specific increase in gamma frequency.

Remarkably, control participants and those with PMDD showed the same dependency of GR frequency on the order of visits (Fig 4). Furthermore, no differences were found between the PMDD and control groups in either the GR frequency or its modulation by the drift rate and MC phase. Thus, contrary to our prediction, frequency of visual gamma oscillations seems to be normal in PMDD.

The effect of the visit order on the GR frequency is interesting by itself. It shows that modulation of gamma frequency by a hormonal status depends on some situational factor. Changes in attention and/or learning are unlikely to explain this result, as neither reaction time, nor percent of omission and commission errors differed between the first and second visit in subjects first investigated during the luteal or follicular phase of the MC (all p's>0.11). On the other hand, according to the STAI questionnaire, the first visit to the MEG laboratory was associated in our participants with significantly higher situational anxiety than the second visit. We, therefore, suggest that a mild acute stress associated with being a study participant for the first time is a probable factor interfering with the hormone levels and contributing to our gamma frequency results. In line with this suggestion, several studies do show increased stress sensitivity in the luteal phase in both women without severe PMS [133, 155, 156] and in women with PMDD [19, 157–161].

The factors associated with the 1st visit led to an increase in GR frequency only if they coincided with the luteal phase of the MC. During this phase, the $\delta GABA_ARs$-mediated inhibition is potentiated by ALLO (see the previous section). On the other hand, the $\delta GABA_ARs$-mediated inhibition is further increased in response to acute stress [162–164]. The increase in $\delta GABA_ARs$-mediated tonic inhibition is expected to dampen excitability of PV+ inhibitory neurons, whose tonic inhibition is exclusively mediated by $\delta GABA_ARs$ [165, 166]. This dampened tonic activity of PV+ inhibitory neurons is expected to decrease GR frequency [66, 167]. Our results, however, strongly indicate the opposite.

Although speculative, the impact of mild acute stress on gamma frequency during the luteal phase may be explained by the paradoxical excitatory effect of GABA-mediated currents on inhibitory neurons. It has been demonstrated that experimentally increased tonic $GABA_ARs$ conductance, mimicking its natural increase during the luteal phase, has a depolarizing effect on adult hippocampal inhibitory neurons [116]. While this excitatory effect, mediated by tonic GABA currents, may not be sufficient to cause a measurable increase in gamma frequency in the luteal phase, mild acute stress, e.g., associated with the first visit, may further increase interneuron excitability. Indeed, acute stress rapidly increases plasma and cortical ALLO levels, which, in turn, leads to upregulation of $\delta GABA_ARs$ [71–73]. That is, when acute stress

coincides with the luteal phase, $\delta GABA_A Rs$ reach their highest density. Apart from the potentiating effect of ALLO, other stress-derived neurosteriods have also been shown to increase the burst firing of inhibitory neurons [72, 168]. The increased excitability of inhibitory neurons due to the additive effects of the luteal phase and mild acute stress could thus explain the increased frequency of gamma oscillations in women whose first visit to the laboratory coincided with the luteal phase of their MC.

Whatever the exact mechanism leading to the luteal phase-specific effect of the first MEG scanning on the visual GR frequency, it did not differ in women with and without PMDD. Overall, the absence of PMDD-specific differences in the frequency of visual GR as well as in its modulation by either excitatory drive (i.e., grating's drift rate), visit order, or MC phase suggests that activity of PV+ interneurons in the visual cortex is not altered in PMDD.

Although GR frequency was not affected in women with PMDD, we did find in this group an altered relationship between GR frequency and strength of GR suppression. In control women, we reproduced the previously reported correlation [64]: greater increase in gamma frequency correlated with greater gamma suppression caused by a strong excitatory drive to the visual cortex. This correlation suggests that in the healthy brain, the inhibitory down-regulation of the E/I balance evidenced by GR power suppression is proportional to excitation of inhibitory PV+ neurons, which define frequency of gamma oscillations (see [64] for thorough discussion). In women with PMDD, GR frequency did not correlate with the GR suppression, and the difference in the correlation coefficients between PMDD and control groups was significant in the luteal phase. This finding suggests that in women with PMDD the presynaptic inhibitory neurons fail to control synchronized activity of overexcited principal cells in an optimal way.

**Perceptual spatial suppression in PMDD.** Contrary to our original hypothesis, which was based on a presumed link between perceptual spatial suppression and cortical inhibition [75, 83], spatial suppression was not reduced in women with PMDD. During the follicular phase, our participants with PMDD tended to have even stronger spatial suppression than control women. This trend is opposite to the effects found in elderly individuals [79, 80] and people with neuropsychiatric disorders that are associated with a disturbed balance between excitation and inhibition [81–84, 99, 169] (but for results similar to ours, see [150]), and is not consistent with an assumed reduction of neural inhibition. One possible reason for this unexpected result is that perceptual spatial suppression is a complex phenomenon, and, as Tadin [75] notes, its association with inhibitory dysfunction is not as straightforward as suggested in some studies cited above. Animal studies of neural surround suppression–a decrease in firing rate of cortical neurons with increasing visual stimulus size–have demonstrated the complexity of this phenomenon, which relies on E/I interactions in the primary visual cortex involving multiple inhibitory cell subtypes, as well as on their top-down regulation by higher-tier cortical areas [77, 170–172]. The balance of neural mechanisms underlying surround suppression is essential for normal perception, while its disruption may lead to an altered perceptual spatial suppression in clinical populations [172].

Given the critical role of top-down influences in surround suppression [170, 173] and perceptual suppression itself [174, 175], our results rather indicate preserved or even elevated top-down control in the asymptomatic follicular phase in women with PMDD. However, the complex hierarchical organization of excitatory-inhibitory interactions underlying perceptual spatial suppression makes it difficult to separate initially elevated inhibition from homeostatic compensation for its deficit (see, e.g., [146]). Interestingly, such a putative compensatory mechanism may also present in patients with migraine, who are thought to have an elevated neural excitability in the visual cortex [176, 177], but at the same time demonstrate increased perceptual spatial suppression [178, 179].

While perceptual spatial suppression did not depend on the MC phase in the control participants, it significantly decreased from the follicular to the luteal phase in women with PMDD (Fig 7B). This decrease may reflect diminished effectiveness of the top-down compensatory influences in women with PMDD during the symptomatic luteal phase. This suggestion is broadly consistent with MRI findings of decreased anterior cingulate cortex control over the amygdala response to negative social stimulation in women with PMDD during the luteal phase compared to the follicular phase ([180]; but see [35, 38]), and with animal studies showing a facilitating effect of anterior cingulate cortex activation on surround inhibition [181].

The atypical regulation of perceptual spatial suppression in women with PMDD is further supported by the lack of the neuro-behavioral correlation found in the control group. Consistent with our previous results in typically developing children [63], visual GR suppression predicted spatial suppression in control women, with stronger gamma suppression corresponding to greater perceptual suppression (see Results section and S3 Table and S2 Text in S3 File for additional analysis). This relationship was absent in women with PMDD, indicating that some atypical compensatory processes may result in apparently normal perceptual spatial suppression in this clinical group.

**Limitations and future direction of research.** One limitation of the present study is the relatively small sample size. Although being comparable to that in the majority of neurophysiological studies into PMS/PMDD pathogenesis, it did not allow us to investigate the subtypes, which likely exist within the PMDD nosological category [105] and which may be associated with distinct neuro-functional deficits. According to the literature, about 50% of women with PMDD have comorbid conditions [85–87]. It would be important to investigate if the constitutively elevated cortical excitability characterizes all women with PMDD, or rather certain patients with specific symptom patterns and/or comorbidities.

Further, our current design contrasts the early-to-mid follicular phase with the mid-to-late luteal phase, while it misses events in the middle of the MC. The inclusion of the ovulatory period where many women with PMDD are already starting to develop adverse symptoms, would help to better understand the relationship between MC-related changes in the E/I balance and PMS onset.

In the future, it would be interesting to model the effects of excitatory drive, acute stress, and PMDD diagnosis on gamma parameters using computer models of gamma generation in the visual cortex.

Finally, our work has been restricted to the occipital cortex, which is not usually involved in the pathogenesis of affective disorders. One cannot rule out the possibility that E/I balance in other, 'core' for PMDD, brain regions will be differently affected by neurosteroids and other factors associated with PMDD.

## Conclusion

Although the main symptoms of PMDD are emotional rather than sensory-related, the neuro-functional abnormalities associated with this disorder are present even in the visual cortex. Our results showing atypical modulation of gamma power as a function of excitatory drive suggest that neuronal excitability is constitutively elevated in women with PMDD, which may be related to the high frequency of comorbid neuropsychiatric conditions in this clinical group. The E/I imbalance in women with PMDD is further exacerbated during the luteal phase, possibly as a result of their atypical sensitivity to neurosteroids. The mechanisms of elevated neural excitability and E/I imbalance in PMDD remain to be elucidated, but these abnormalities are rather unlikely to be caused by deficits in inhibitory neurons. Indeed, the peak frequency of oscillatory gamma response, which is primarily regulated by tonic excitability of

parvalbumin-containing interneurons, was normal in PMDD. Moreover, women with PMDD and control women demonstrated the same luteal phase-specific effect of 'the first scan' on the gamma peak frequency. The presence of normal or even slightly increased (in the follicular phase) perceptual spatial suppression in PMDD also indicates that there is no major deficit in activity of inhibitory neurons, at least in the visual cortex. Therefore, our results suggest that the changes in neural E/I balance in the visual cortex in women with PMDD are most likely explained by increased tonic excitability of the principal cells and/or impaired regulation of their excitability at the synaptic levels. Overall, our results contribute to elucidating the mechanisms of PMDD and provide new insights into the functional correlates of the visual gamma rhythm.

## Supporting information

**S1 Fig.** Violin plots for gamma response (GR) power (upper panel) and GR frequency (lower panel). Note that the group differences were not significant for all of these GR parameters (t-test, all p's>0.12; uncorrected for multiple comparisons).
(DOCX)

**S2 Fig. Violin plots of gamma response (GR) suppression and GR facilitation scores.**
(DOCX)

**S1 File.**
(DOCX)

**S2 File. Results of bipolar disorder and depression questionnaires in women with PMDD.**
(DOCX)

**S3 File. Gamma suppression slope and perceptual spatial suppression.**
(DOCX)

**S1 Table. Partial Spearman's correlations between steroid hormone levels (estradiol, progesterone) and gamma response (GR) power and frequency adjusted for age in the two groups of participants (control, PMDD).** Correlations with p<0.05 (uncorrected for multiple comparisons) are highlighted in bold.
(DOCX)

**S1 Data. The data used for statistical analysis.**
(XLSX)

## Acknowledgments

We sincerely thank all of the women who participated in this study.

The study was conducted at the unique research facility "Center for Neurocognitive Research (MEG-Center)" of MSUPE.

## Author Contributions

**Conceptualization:** Elena V. Orekhova, Tatiana A. Stroganova.

**Data curation:** Viktoriya O. Manyukhina, Elena V. Orekhova, Andrey O. Prokofyev, Tatiana S. Obukhova.

**Formal analysis:** Viktoriya O. Manyukhina, Elena V. Orekhova.

**Funding acquisition:** Elena V. Orekhova, Tatiana A. Stroganova.

**Investigation:** Viktoriya O. Manyukhina, Tatiana S. Obukhova.

**Methodology:** Elena V. Orekhova, Tatiana A. Stroganova.

**Project administration:** Elena V. Orekhova, Tatiana A. Stroganova.

**Resources:** Andrey O. Prokofyev.

**Software:** Tatiana S. Obukhova.

**Supervision:** Elena V. Orekhova, Tatiana A. Stroganova.

**Validation:** Elena V. Orekhova, Andrey O. Prokofyev.

**Visualization:** Viktoriya O. Manyukhina, Elena V. Orekhova.

**Writing – original draft:** Viktoriya O. Manyukhina, Elena V. Orekhova.

**Writing – review & editing:** Viktoriya O. Manyukhina, Elena V. Orekhova, Andrey O. Prokofyev, Tatiana S. Obukhova, Tatiana A. Stroganova.

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
