## [Decision Letter · Decision Letter 0]

28 Oct 2022

PONE-D-22-26872Altered visual cortex excitability in premenstrual dysphoric disorder: evidence from magnetoencephalographic gamma oscillations and perceptual suppressionPLOS ONE

Dear Dr. Orekhova,

Thank you for submitting your study.

This is a very interesting and worthwhile proposal.

Also you will notice that reviewers enjoyed your study, but required some amendments.

**Please respond to all comments AND highlight the tracked changes.**

We look forward to receiving your revised manuscript.

Kind regards,

Thiago P. Fernandes, PhD

Academic Editor

PLOS ONE

Reviewers' comments:

Reviewer's Responses to Questions

**Comments to the Author**

1. Is the manuscript technically sound, and do the data support the conclusions?

Reviewer #1: Yes

Reviewer #2: Partly

Reviewer #3: Yes

2. Has the statistical analysis been performed appropriately and rigorously? 

Reviewer #1: Yes

Reviewer #2: Yes

Reviewer #3: Yes

3. Have the authors made all data underlying the findings in their manuscript fully available?

Reviewer #1: Yes

Reviewer #2: No

Reviewer #3: Yes

4. Is the manuscript presented in an intelligible fashion and written in standard English?

Reviewer #1: Yes

Reviewer #2: Yes

Reviewer #3: Yes

5. Review Comments to the Author

Reviewer #1: The authors of this manuscript focus on a timely and interesting research question of how the menstrual-related neurosteroids influence excitation and inhibition in sensory cortex, in both the normal menstrual cycle and in the context of premenstrual dysphoric disorder (PMDD). They focus on the visual cortex, with behavioural measures including induced MEG gamma oscillations, and the behavioural measure of surround suppression.

The neural and behavioural metrics, and paradigms in which these are tested, are well-characterised, with considerable scientific underpinnings known. As such, they make excellent choices for addressing the authors' hypotheses. On that note, the hypotheses are clear and well-articulated.

An interesting pattern of results is seen, with the notable differences being:

PMDD is associated with:

Reduced luteal phase progesterone levels

Slower luteal phase reaction time

Increased gamma power facilitation from static to slow drifting gratings in both menstrual phases

Decreased gamma power suppression from slow to medium drifting gratings in the luteal phase only

Decreased perceptual surround suppression from follicular to luteal phase

Findings are interpreted as PMDD being associated with:

A luteal phase specific deficit in cortical inhibition

A phase-general increase in cortical excitation

Overall comments:

In short, I think this is an excellent scientific manuscript throughout. The research area, aims and hypotheses are interesting and novel. The overall paradigm and approach is well-chosen, and implemented with rigorous attention to methodological detail and elimination of potential confounds. MEG data analysis is soundly conducted, and in line with best practice.

The results are interesting, and likely to be of broad relevance across a large numbers of basic and clinical neuroscience. The interpretation given by the authors is well-argued and seems very plausible, and avoids excessive speculation.

I have only a few minor comments, and do not feel that further peer review is necessary to check the revisions arising from these:

Line 180: ‘liability’ should be ‘lability’

Line 196: ‘Besides’ should be ‘Additionally’

Line 539: ‘menstrual’ should be ‘menstrual cycle’

Line 673: it is stated that perceptual surround suppression was not affected in PMDD, but lines 620-627 in the Results section state and discuss that a significant difference was seen. Please clarify.

Reviewer #2: Manyukhina and colleagues present a study of visual functioning in women with premenstrual dysphoric disorder (PMDD) versus controls. They hypothesized that the balance of excitatory and inhibitory (E/I) functions would be altered in PMDD participants due to disrupted hormone sensitivity. They sought to test this hypothesis using two visual paradigms: 1) gamma-band responses (GR) to static and moving gratings measured with MEG, and 2) spatial suppression of motion discrimination (duration thresholds), measured in a behavioral task.

Overall, this study is very thorough. The authors paid careful attention to potential confounds (visit day, age) and provided good justification for the statistical tests that were used. The question of whether E/I imbalance disrupts visual perception in PMDD is an interesting one. However, there are a number of issues that limit my enthusiasm for the current manuscript.

Major points:

• Throughout the manuscript, the authors assert that their measure of spatial suppression is explained by inhibition in visual cortex, but this point appears somewhat controversial within the literature, and empirical support is equivocal. In contrast to the work that is cited, a number of studies in both humans and animal models have failed to find a strong link between visual center-surround suppression and inhibitory functioning (Liu et al., 2018; Ozeki et al., 2009; Ozeki et al., 2004; Read et al., 2015; Sato et al., 2016; Schach et al., 2021; Schallmo et al., 2018; Schallmo et al., 2020). Some of these have proposed alternative models, wherein withdrawal of recurrent excitation plays a primary role in the suppression phenomenon, with inhibition playing a secondary or supporting role. To some extent, this seems like an issue about the semantic distinction between ‘inhibition’ (i.e., GABAergic hyperpolarization) and ‘suppression’ (i.e., reduced perceptual functioning). Although the authors’ theory regarding the link between motion suppression and inhibition is interesting, it would be prudent to clarify that this is an area of active research, rather than a settled point in the literature.

• The issue of co-morbidity between PMDD and bipolar disorder or depression is important, as visual disruptions, including in center-surround tasks, have been observed in these conditions (Golomb et al., 2009; Miller et al., 2003; Ngo et al., 2011; Norton et al., 2016; Schallmo et al., 2015; Yang et al., 2013). This topic is touched on briefly in the limitations section, but is not fully addressed. How might the authors differentiate between specific effects of PMDD versus effects of co-morbid mood disorders?

• Regarding the GR suppression metric, I am concerned that expressing this value as a ratio distorts the pattern of results. It appears that the PMDD group has higher GR power in both the slow and medium conditions vs. controls, as supported by the significant group * velocity interaction. This overall difference could induce a spurious difference between these two groups when a ratio metric is used. Instead, a subtraction metric appears warranted. See (Curran-Everett, 2013). This comment applies equally to GR facilitation.

• According to the text in the Results, the authors calculate their suppression index in a way that is not standard in the field; the standard metric is the log of the difference in threshold values (Tadin, 2003; Foss-Feig, 2013), rather than the ratio. Using the current metric makes it difficult to compare the results of this study to the wider literature. However, the methods state that the standard difference metric was used. Please clarify.

• The manuscript is overly-long in general, but frustratingly, appears to leave out important details and justification in places (some examples are provided in the Minor points below). I believe the manuscript would benefit from greater use of supplemental materials. Some tables and analyses (e.g., sections 3.1 – 3.3) are less relevant to the main hypotheses than others and could be briefly summarized in the main text, with details in the supplemental text.

Minor points:

• Line 38 – please clarify that suppression refers to GR power, not perception.

• Lines 106 & 110 – please specify GR power rather than frequency.

• Please provide an explanation for why controls did not also complete the C-PASS.

• Which ethics body specifically reviewed the study, and how were participants compensated for their time?

• Why were 26 vertices used, and are these in the same position for all participants? Were they required to be contiguous?

• Why not include motion discrimination data from the 2.5 degree size condition? No justification is given, and these data might be expected to aid the interpretation of the pattern of results, as the pattern of spatial suppression across stimulus sizes is well established in healthy adults.

• Data points from individual participants are shown in Figures 2 and 3, which is appreciated, but this is not the case for later figures.

• It appears the correlations in Table 3 were not corrected for multiple comparisons. This should be done, or at least included in addition to uncorrected p-values, given the large numbers of tests reported (16).

• Line 641 – The authors say "In control women, the percent of GR suppression from the ‘slow’ to the ‘medium’ condition (Fig 6c) correlated with the SSI" but the correlation is only significant for the follicular phase, please clarify.

• Line 647 – Is there a significant group difference in the GR suppression and SSI correlations themselves?

References

Curran-Everett, D. (2013). Explorations in statistics: the analysis of ratios and normalized data. Advances in Physiology Education, 37, 213-219.

Golomb, J. D., McDavitt, J. R. B., Ruf, B. M., Chen, J. I., Saricicek, A., Maloney, K. H., Hu, J., Chun, M. M., & Bhagwagar, Z. (2009). Enhanced visual motion perception in major depressive disorder. The Journal of Neuroscience, 29(28), 9072-9077.

Liu, L., Miller, K. D., & Pack, C. C. (2018). A unifying motif for spatial and directional surround suppression. The Journal of Neuroscience, 38(4), 989-999.

Miller, S. M., Gynther, B. D., Heslop, K. R., Liu, G. B., Mitchell, P. B., Ngo, T. T., Pettigrew, J. D., & Geffen, L. B. (2003). Slow binocular rivalry in bipolar disorder. Psychological Medicine, 33(4), 683-692.

Ngo, T. T., Mitchell, P. B., Martin, N. G., & Miller, S. M. (2011). Psychiatric and genetic studies of binocular rivalry: an endophenotype for bipolar disorder? Acta Neuropsychiatrica, 23(1), 37-42.

Norton, D. J., McBain, R. K., Pizzagalli, D. A., Cronin-Golomb, A., & Chen, Y. (2016). Dysregulation of visual motion inhibition in major depression. Psychiatry Research, 240, 214-221.

Ozeki, H., Finn, I. M., Schaffer, E. S., Miller, K. D., & Ferster, D. (2009). Inhibitory stabilization of the cortical network underlies visual surround suppression. Neuron, 62, 578-592.

Ozeki, H., Sadakane, O., Akasaki, T., Naito, T., Shimegi, S., & Sato, H. (2004). Relationship between excitation and inhibition underlying size tuning and contextual response modulation in the cat primary visual cortex. The Journal of Neuroscience, 24(6), 1428-1438.

Read, J. C. A., Georgiou, R., Brash, C., Yazdani, P., Whittaker, R., Trevelyan, A. J., & Serrano-Pedraza, I. (2015). Moderate acute alcohol intoxication has minimal effect on surround suppression measured with a motion direction discrimination task. Journal of Vision, 15, 5-5.

Sato, T. K., Haider, B., Haüsser, M., & Carandini, M. (2016). An excitatory basis for divisive normalization in visual cortex. Nature neuroscience, 1-3.

Schach, S., Surges, R., & Helmstaedter, C. (2021). Visual surround suppression in people with epilepsy correlates with attentional-executive functioning, but not with epilepsy or seizure types. Epilepsy & Behavior, 121, 108080.

Schallmo, M.-P., Kale, A. M., Millin, R., Flevaris, A. V., Brkanac, Z., Edden, R. A. E., Bernier, R. A., & Murray, S. O. (2018). Suppression and facilitation of human neural responses. eLife, 7, e30334.

Schallmo, M.-P., Kolodny, T., Kale, A. M., Millin, R., Flevaris, A. V., Edden, R. A. E., Gerdts, J., Bernier, R. A., & Murray, S. O. (2020). Weaker neural suppression in autism. Nature communications, 11(1), 2675.

Schallmo, M.-P., Sponheim, S. R., & Olman, C. A. (2015). Reduced contextual effects on visual contrast perception in schizophrenia and bipolar affective disorder. Psychological Medicine, 45(16), 3527-3537.

Yang, E., Tadin, D., Glasser, D. M., Hong, S. W., Blake, R., & Park, S. (2013). Visual context processing in bipolar disorder: A comparison with schizophrenia. Frontiers in Psychology, 4(569), 1-12.

Reviewer #3: The authors investigate the difference in the E-I balance between post-menstrual dysphoric disorder (PMDD) subjects and healthy controls using the changes in gamma-band power to the drifting annular gratings of varying velocity and perceptual suppression task. The way gamma power and frequency vary with drift rate has been proposed to be an indicator of E/I balance. Therefore, the authors hypothesize that during the symptomatically intensive luteal phase, the differences in E-I balance is prominent and should be reflected in the gamma frequency, suppression of gamma power with drift rate and performance of perceptual spatial suppression task. While the authors did not find the expected results, they did find some differences between PMDD and control subjects for some drift rates.

First, I think irrespective of the results, the study has been done very well. The study design is thorough, statistics are sound, literature survey is thorough and excellent. It is also clear that the authors have worked very hard to write a strong Discussion section (probably based on comments in previous submissions) to address a variety of questions that could have come up while reading the paper.

My only major comment is to perhaps tone down the Abstract a little bit and explicitly state that the main effect that they were looking for is not present (i.e., it is a null result as far as the main prediction is concerned). This does not take anything away from the work. The reasons provided in the Discussion are thorough and could lead to more refined studies in future. Overall, I endorse the publication of this manuscript.

Minor:

1) Line 75-77: There is an error in qualifying the concentration of GABAA receptors in visual cortex. Authors state “Visual cortex displays the highest concentration of GABAA and GABAB receptors among other brain areas in humans” but the figure 7 and figure 8 of the cited reference [45] show the highest concentration of GABAA in the auditory cortex.

2) Figure 1: Does MEG task panel in the figure depict a single trial? If so, it doesn’t match the description of the task in the methods. In the method (line 225), it is described as “Each new trial started with a fixed 1.2 s prestimulus interval…” but in the figure the trial seems to start with the stimulus followed by a prestimulus period. Also, in the figure it seems as if two stimuli are shown in a trial but it’s not the case according to the methods. An arrow indicating the direction of time can be helpful here.

3) The psychophysical task is not very clear from the figure or from the methods. Little more details on the task, especially the structure of a trial in both methods and figure will be helpful. I was unable to find reference [98] online, which is cited for the details of the task. From the figure it seems as if stimuli of different sizes are presented in a trial and response is sought only for the one presented at the end? Is that the case?

Reference [98]:

Manyukhina VO, Rostovtseva EN, Prokofyev AO, Stroganova TA, Orekhova E V. Menstrual cycle-linked changes in neural oscillations measured by MEG may provide biomarker for premenstrual dysphoric disorder. 2020;(2010):2020.

4) Line 618: There seems to be a mistake in the SSI formula. The ratio of the threshold should have been large/small.

5) Line 645: Three is misspelled as tree.

6) Line 818-820: Authors argue that GR facilitation in PMDD is unlikely to depend on MC phase related fluctuations in neurosteroids and the effect of neurosteroids on GABAA mediated inhibition because it enhanced in both the phases. However, this alone is not sufficient to rule out the effect of neurosteroid on GABAA mediated tonic inhibition in case there is a tolerance to ALLO in the luteal phase as the authors hypothesize in section 4.2.1. The evidence for unlikely dependence of facilitation on the effects of neurosteroid on GABAA receptor comes from the control group where the facilitation is maintained despite the absence of the tolerance to ALLO.

7) Table 3: Correlations were significant for control women but not for PMDD subjects. Any reason why?

8) Line 256: Is it ‘Humming’ or Hamming window?

6. PLOS authors have the option to publish the peer review history of their article (what does this mean?). If published, this will include your full peer review and any attached files.

Reviewer #1: **Yes: **William Sedley

Reviewer #2: No

Reviewer #3: No

---

## [Author Response · Author response to Decision Letter 0]

23 Nov 2022

Please, see the attached file with our responses to reviewers' comments.

---

## [Decision Letter · Decision Letter 1]

19 Dec 2022

Altered visual cortex excitability in premenstrual dysphoric disorder: evidence from magnetoencephalographic gamma oscillations and perceptual suppression

PONE-D-22-26872R1

Dear Dr. Orekhova,

We’re pleased to inform you that your manuscript has been judged scientifically suitable for publication and will be formally accepted for publication once it meets all outstanding technical requirements.

Kind regards,

Thiago P. Fernandes, PhD

Academic Editor

PLOS ONE

Additional Editor Comments (optional):

Please check your data disposition acorrding to the PLOS policies, also check your references and a few issues on grammar when in typesetting. Thank you for your valuable subsmission. Hope you can consider PLOS again.

Reviewers' comments:

Reviewer's Responses to Questions

**Comments to the Author**

1. If the authors have adequately addressed your comments raised in a previous round of review and you feel that this manuscript is now acceptable for publication, you may indicate that here to bypass the “Comments to the Author” section, enter your conflict of interest statement in the “Confidential to Editor” section, and submit your "Accept" recommendation.

Reviewer #1: All comments have been addressed

Reviewer #2: All comments have been addressed

2. Is the manuscript technically sound, and do the data support the conclusions?

Reviewer #1: Yes

Reviewer #2: Yes

3. Has the statistical analysis been performed appropriately and rigorously? 

Reviewer #1: Yes

Reviewer #2: Yes

4. Have the authors made all data underlying the findings in their manuscript fully available?

Reviewer #1: Yes

Reviewer #2: Yes

5. Is the manuscript presented in an intelligible fashion and written in standard English?

Reviewer #1: Yes

Reviewer #2: Yes

6. Review Comments to the Author

Reviewer #1: From my perspective, and my points raised, I am satisfied that this manuscript is ready for publication.

That said, my comments were already minor and limited, and the bigger question is whether the other two reviewers are satisfied with the responses to their comments and questions. At a glance, these seem to have been satisfactorily addressed, but the other two reviewers will be in a better position to judge this than I am.

Reviewer #2: The authors have made an admirable effort to address the concerns raised by myself and the other reviewers. I thank them, and I have no further comments.

7. PLOS authors have the option to publish the peer review history of their article (what does this mean?). If published, this will include your full peer review and any attached files.

Reviewer #1: **Yes: **William Sedley

Reviewer #2: No

---

## [Editor Report · Acceptance letter]

22 Dec 2022

PONE-D-22-26872R1 

Altered visual cortex excitability in premenstrual dysphoric disorder: evidence from magnetoencephalographic gamma oscillations and perceptual suppression 

Dear Dr. Orekhova:

I'm pleased to inform you that your manuscript has been deemed suitable for publication in PLOS ONE. Congratulations! Your manuscript is now with our production department. 

Kind regards, 

on behalf of

Dr. Thiago P. Fernandes 

Academic Editor

PLOS ONE